**Improving PM$_{2.5}$ forecast over China by the joint adjustment of initial conditions**
**and source emissions with an ensemble Kalman filter**
Zhen Peng[1,2], Zhiquan Liu[2], Dan Chen[2], Junmei Ban[2]
1 School of Atmospheric Sciences, Nanjing University, Nanjing, China
2 National Center for Atmospheric Research, Boulder, Colorado, USA
**Abstract.** In an attempt to improve the forecasting of atmospheric aerosols, the
ensemble square root filter algorithm was extended to simultaneously optimize the
chemical initial conditions and emission input. The forecast model, which was
expanded by combining the Weather Research and Forecasting with Chemistry
(WRF-Chem) model and a forecast model of emission scaling factors, generated both
chemical concentration fields and emission scaling factors. The forecast model of
emission scaling factors was developed by using the ensemble concentration ratios of
the WRF-Chem forecast chemical concentrations and also the time smoothing
operator. Hourly surface fine particulate matter (PM$_{2.5}$) observations were assimilated
in this system over China from 5 to 16 October 2014. A series of 48-h forecasts were
then carried out with the optimized initial conditions and emissions on each day at
0000 UTC and a control experiment was performed without data assimilation. Besides,
we also performed an experiment of pure assimilation chemical ICs and the
corresponding 48-h forecasts experiment for comparison. The results showed that the
forecasts with the optimized initial conditions and emissions typically outperformed
those from the control experiment. In the Yangtze River delta (YRD) and the Pearl
River delta (PRD) regions, large reduction of the Root Mean Square Errors (RMSEs)
was obtained for almost the entire 48-h forecast range attributed to assimilation.
Especially, the relative reduction in RMSE due to assimilation was about 37.5% at
nighttime when WRF-Chem performed comparatively worse. In the Beijing–Tianjin–
Hebei (JJJ) region, relatively smaller improvements were achieved in the first 24-h
forecast but then no improvements were achieved afterwards. Comparing to the
forecasts with only the optimized ICs, the forecasts with the joint adjustment were
always much better during the night in the PRD and YRD regions. However, they
were very similar during daytime in both regions. Also, they performed similarly for
almost the entire 48-h forecast range in the JJJ region.
**1. Introduction**
Aerosol prediction by regional air quality model in heavy polluted regions is
challenging due to many factors. In addition to the deficiency of chemistries, the
uncertainties of primary and precursor emissions and the initial conditions (ICs) also
limit the forecast accuracy. Data assimilation (DA), which is used to improve the ICs
of aerosols and to optimize data on aerosol emissions, has been shown to be one of
the most effective ways to improve the forecasting of aerosol pollution.
From the perspective of reducing the uncertainties in the ICs for aerosols, recent
efforts have focused on assimilating aerosol observations using optimal interpolation
(Collins et al., 2001; Yu et al., 2003; Adhikary et al., 2008; Tombette et al., 2009; Lee
et al., 2013) or variational (Kahnert, 2008; Zhang et al., 2008; Benedetti et al., 2009;
Pagowski et al., 2010; Liu et al., 2011; Schwartz et al., 2012; Li et al., 2013; Jiang et
al., 2013; Saide et al., 2013) DA algorithms. Ensemble-based DA algorithms, such as
the ensemble Kalman filter (EnKF) (Sekiyama et al., 2010; Schutgens et al., 2010a,
2010b; Pagowski and Grell, 2012; Dai et al., 2014; Rubin et al., 2016; Ying, X.M., et
al., 2016; Yumimoto et al., 2016) and the hybrid variational-ensemble DA approach
(Schwartz et al., 2014) have also been applied to aerosol predictions. All these studies
have shown that DA is one of the most effective ways of improving aerosol
forecasting through assimilating aerosol observations from multiple sources (e.g.
ground-based observations and satellite measurements) to update the chemical ICs.
Numerous studies have used DA approaches to estimate or improve source
emissions. The EnKF is one of the most popular DA algorithms used to improve
estimates of aerosols and gas-phase emissions, such as $NO_x$, volatile organic
compounds, and $SO_2$ (van Loon et al., 2000; Heemink and Segers, 2002; Zhang et al.,
2005; Barbu et al., 2009; Sekiyama et al., 2010; Huneeus et al., 2012; Schutgens et al.,
2012; Huneeus et al., 2012, 2013; Miyazaki et al., 2014). Variational DA algorithms
have also been applied to constrain emissions of air pollution, such as black carbon,
organic carbon, dust, NH$_3$, SO$_x$ and NO$_x$ (Hakami et al., 2005; Elbern et al., 2007;
Henze et al., 2007, 2009; Yumimoto et al., 2007, 2008; Dubovik et al., 2008; Wang et
al., 2012; Guerrette and Henze, 2015). These studies have indicated that DA can
efficiently reduce the uncertainty in the emission inventories and lead to
improvements in the forecasting of air quality (Mijling and van der A, 2012).
The optimization of chemical ICs and pollution emissions can improve aerosol
forecasts and therefore further improvements are likely to be achieved by
simultaneously optimizing the chemical ICs and emissions. Tang et al. (2011)
reported that the simultaneous adjustment of the ICs of O$_3$, NO$_x$ and volatile organic
compounds and the emissions of NO$_x$ and volatile organic compounds produced
overall better performance in both the 1-h and 24-h ozone forecasts than the
adjustment of pure ICs or emissions. Miyazaki et al. (2012) reported that the
simultaneous adjustment of emissions and concentrations is a powerful approach to
correcting the tropospheric ozone budget and profile analyses.
We developed a system to adjust the chemical ICs and source emissions jointly
within an EnKF system coupled to the Weather Research and Forecasting with
Chemistry (WRF-Chem) model (Grell et al., 2005). We then applied this system to
assimilate hourly surface PM$_{2.5}$ measurements over China in early October 2014.
The remainder of the paper is organized as follows. Section 2 describes this DA
system in detail and Section 3 describes the PM$_{2.5}$ observations. Then the
experimental designs are introduced in Section 4. Finally, the surface PM$_{2.5}$
observations assimilation results are presented in section 5 before concluding in
section 6.

**2. Methodology**
**2.1 Forecast model**
For a chemical model like WRF-Chem, the emissions are the model forcing (or
boundary condition), rather than model states. Therefore, a forecasting model, **M**,
was developed to forecast the emission scaling factors (representing emissions) as
well as the aerosol concentrations. This model combines the WRF-Chem model and
the forecast model of emission scaling factors.

2.1.1 WRF-Chem model
Version 3.6.1 of the WRF-Chem model (Grell et al., 2005) was used to forecast the
aerosol and chemical species. WRF-Chem is an online model with the fully coupled
chemical and meteorological components.

Most of the WRF-Chem settings were the same as those reported in Liu et al.

(2011): the Goddard Chemistry Aerosol Radiation and Transport (GOCART) aerosol
scheme coupled with the Regional Atmospheric Chemistry Mechanism for gaseous
chemical mechanisms; the WRF single-moment five-class microphysics scheme; the
Rapid Radiative Transfer Model longwave and Goddard shortwave radiation schemes;
the Yonsei University (YSU) boundary layer scheme; the Noah land surface model;
and the Grell-3D cumulus parameterization. For the GOCART aerosol scheme, the
aerosol species include 14 defined aerosol species and a $15^{th}$ variable representing
unspectiated aerosol contributions ($P_{25}$). The 14 defined aerosol species are sulfate,
hydrophobic and hydrophilic organic carbon ( $OC_1$ and $OC_2$ , respectively),
hydrophobic and hydrophilic black carbon ($BC_1$ and $BC_2$, respectively), dust in five
particle size bins (effective radii of 0.5, 1.4, 2.4, 4.5 and 8.0 μm; referred to as $D_1$,
$D_2$, $D_3$, $D_4$ and $D_5$, respectively) and sea salt in four particle size bins (effective
radii of 0.3, 1.0, 3.25 and 7.5 μm for dry air; referred to as $S_1$, $S_2$, $S_3$ and $S_4$,
respectively).

Figure 1 illustrates the model computational domain. It has 120*120 horizontal

grid scales at a 40.5 km spacing by the lambert conform map projection centered at
(35 °N, 105 °E). There are 57 vertical levels with the model top at 10 hPa, about 12
layers within the planetary boundary layer (among them the lowest 8 layers were
under 500 m), and the first layer centered at ~12 m.

With respect to the emissions, the hourly prior anthropogenic emissions were

based on the monthly regional emission inventory in Asia (Zhang et al., 2009) for the
year 2006 interpolated to the model grid. The power generator emissions were
interpolated for the lowest eight vertical levels (Woo et al., 2003; de meij et al., 2006;
Wang et al., 2010). Other anthropogenic emissions were assigned totally to the 1[st]
level. Emissions are very small above 500 m for all pollutants. In order to keep
objective for the prior anthropogenic emissions, no time variation was added. Thus,
the hourly prior anthropogenic emissions were constant. The biogenic (Guenther et al.,
1995), dust (Ginoux et al., 2001), dimethylsulfide and sea salt emissions (Chin et al.,
2000, 2002) were calculated online.

2.1.2 Forecast model of scaling factors
As no suitable dynamic model was available to forecast the emission scaling factors, a
persistence forecasting operator served as the forecast model for the scaling factors,
similar to the method used by Peng et al. (2015) for $CO_2$ emission inversion. Figure
2a shows the flowchart for the persistence forecasting operator $\mathbf{M}_{SF}$.

If the ensemble members of the updated chemical fields $\mathbf{C}^a_{i,t-1}$ (the subscript $i$

refers to the $i$th ensemble member, the superscript a refers to the analysis, and $t$
refers to the time) and the forecast emissions $\mathbf{E}^f_{i,t-1}$ (the superscript f refers to the
forecast) in the previous assimilation cycle are known, then the chemical fields $\mathbf{C}^f_{i,t}$
at time $t$ can be generated via WRF-Chem (Figure 2b). In the actual process, $\mathbf{C}^f_{i,t}$
were available in the previous assimilation cycle, so we did not need to perform the
ensemble forecasts again. A dotted box was used in Figure 2a to indicate that the
ensemble forecasts were not performed in real process. The ensemble concentration
ratios $\kappa_{i,t}, (i = 1, ..., N)$ are then calculated using
$$\kappa_{i,t} = \frac{\mathbf{C}^f_{i,t}}{\overline{\mathbf{C}^f_t}}, (i = 1, ..., N), (1)$$
where $\overline{\mathbf{C}^f_t} = \frac{1}{N}\sum_{i=1}^N \mathbf{C}^f_{i,t}$ is the ensemble mean of the forecast. The ensemble mean of
$\kappa_{i,t}$ is,
$$\overline{\kappa_t} = \frac{1}{N}\sum_{i=1}^N \kappa_{i,t} = \frac{1}{N}\sum_{i=1}^N \mathbf{C}^f_{i,t}/\overline{\mathbf{C}^f_t} = 1, (2)$$
so $\kappa_{i,t}$ are numbers distributed around 1 and with ensemble mean values of 1.

The ensemble spreads of $\kappa_{i,t}, (i = 1, ..., N)$ may be small and therefore

covariance inflation is used to maintain them at a certain level:
$$(\kappa_{i,t})_{\mathrm{inf}} = \beta\left(\kappa_{i,t} - \overline{\kappa_t}\right) + \overline{\kappa_t}, (i = 1, \ldots, N), (3)$$
In Peng et al. (2015), the $CO_2$ DA system worked comparatively well when the
ensemble spread of $\lambda_{i,t}^{a}$ ranged from 0.05 to 1.25 for $\beta = 60$, 70, 75, 80. The
assimilated CO2 fluxes deviated markedly from the "true" CO2 fluxes when the
ensemble spread of $\lambda_{i,t}^{a}$ were too small for $\beta = 10$, 50 or when the ensemble spread
of $\lambda_{i,t}^{a}$ were too large for $\beta = 100$. Therefore, in this work, $\beta = 1.5$ was chosen to
make ensure the ensemble spread of $(\kappa_{i,t})_{\mathrm{inf}}$ ranged from 0.1 to 1.25. Same as $\kappa_{i,t}$,
the ensemble mean values of $(\kappa_{i,t})_{\mathrm{inf}}$ are 1. It is noted that perhaps there are very
few negative values for $(\kappa_{i,t})_{\mathrm{inf}}$ after inflation. A quality control procedure is
performed for $(\kappa_{i,t})_{\mathrm{inf}}$ before further appliance. All these negative data were set as
0.001 in this work. There was no special reason to set them as 0.001. It is also fine to
set them as 0. Then $(\kappa_{i,t})_{\mathrm{inf}}$ were re-centered to ensure the ensemble mean values of
$(\kappa_{i,t})_{\mathrm{inf}}$ were all 1.
As the concentrations were closely related to the emissions both locally and in
the upwind regions and there is no suitable dynamic model available to forecast the
emission scaling factors, the inflated concentration ratios $(\kappa_{i,t})_{\mathrm{inf}}$ serve as the prior
emission scaling factors $\lambda_{i,t}^{p}$:
$$\lambda_{i,t}^{p} = (\kappa_{i,t})_{\mathrm{inf}}, (i = 1, \ldots, N), (4)$$
The above equation is not supported according to the mass conservation equation
but just for the purpose to generate the ensemble emissions. Same as $(\kappa_{i,t})_{\mathrm{inf}}$, $\lambda_{i,t}^{p}$
are numbers distributed around 1. From the perspective of generating the ensemble
emissions, they can play the same role as other data, such as the random numbers
created by using the standard normal distribution function. However, there are
correlations among the grid-points of $(\kappa_{i,t})_{\mathrm{inf}}$ because $(\kappa_{i,t})_{\mathrm{inf}}$ are calculated
through a short-term forecast of WRF-Chem. Thus, $\lambda_{i,t}^{p}$ have the same correlations
as $(\kappa_{i,t})_{\mathrm{inf}}$. While, the random numbers are totally different. There are no
correlations unless they are generated under certain correlations.
To incorporate the useful information from the previous times, the previous DA
cycles' analysis scaling factors, $\lambda^{a}_{i,t-M+1}$, $\cdots$, $\lambda^{a}_{i,t-2}$, $\lambda^{a}_{i,t-1}$ and the prior scaling
factor $\lambda^{p}_{i,t}$ were used to estimate $\lambda^{f}_{i,t}$ by the time smooth operator; namely,
$\lambda^{f}_{i,t} = \frac{1}{M}\left(\sum_{j=t-M+1}^{t-1}\lambda^{a}_{i,j} + \lambda^{p}_{i,t}\right), (\, i = 1, \dots, N, j = t - M + 1, \dots, t - 1), (5)$
Here, $M$ is the time window of the smooth operator. In this study, a value of $M = 4$
(hours) was chosen. According to the smooth operator, the ensemble mean values of
$\lambda^{f}_{i,t}$ depend on the ensemble mean of $\lambda^{a}_{i,t-M+1}$, $\cdots$, $\lambda^{a}_{i,t-2}$, $\lambda^{a}_{i,t-1}$, $\lambda^{p}_{i,t}$, where the
ensemble means of $\lambda^{p}_{i,t}$ are all 1. After multiple iterations, the smooth operator can
give comparatively good estimation for $\lambda^{f}_{i,t}$ since anthropogenic emissions are stable
at a certain time scale (Mijling et al., 2012). It is a compromise between prescribed
prior emissions and letting the system propagate all observation information from one
step to the next without any guidance (Peters et al., 2007), for the case $M = 4$.
The ensemble members of the emissions were calculated according to
$$\mathbf{E}_{i,t} = \lambda_{i,t}\mathbf{E}^{p}_{t}, (i = 1, \dots, N), (6)$$
where $\mathbf{E}_{i,t}$ is the $i$th ensemble member of the emissions for each grid at time $t$, $\lambda_{i,t}$
represents the scaling factors and $\mathbf{E}^{p}_{t}$ is the prescribed emission, which can be
obtained from the emission inventories. It is noted that the correlations among the
grid-points of the prior emissions depend on $\lambda^{p}_{i,t}$. These correlations may deviate far
from the truth but we have no other suitable substitute. However, the correlations
among the grid-points of the forecast emissions should be more or less close to the
truth due to the appliance of the smooth operator after multiple iterations.
It is noted although the method is very similar to that used by Peters et al. (2007)
and Peng et al. (2015) for $CO_2$ emission inversion, it is still of novelty for applications
in aerosol anthropogenic emissions. In Peters et al. (2007), $\lambda^{p}_{i,t}$ were all 1. And only
natural $CO_2$ emissions (i.e., biospheric and oceanic emissions) were assimilated at the
ecological scale due to the 'signal-to-noise' problem. Thus, the uncertainty of
anthropogenic and other $CO_2$ emissions were ignored. Besides, the framework is more
advanced compared to our previous work. In Peng et al. (2015), in order to generate
$\lambda_{i,t}^{p}$, a set of ensemble forecasts were performed from time $t$ to $t+1$ to produce the $CO_2$
concentration fields, forced by the prescribed net $CO_2$ surface fluxes with the previous
assimilated concentration fields as initial conditions. That means that the ensemble
forecast were performed twice in that DA system and it was time consuming.
However, in order to save computing time, we used the chemical fields $\mathbf{C}_{i,t}^{f}$ available
in the previous assimilation cycle to calculate $\lambda_{i,t}^{p}$ in this work. Thus, WRF-Chem
runs to forecast only once during a DA cycle.

**2.2 Ensemble square root filter**
The ensemble square root filter (EnSRF) algorithm was introduced by Whitaker
and Hamill (2002) and its expansion to analyzing aerosol ICs was described by
Schwartz et al. (2014). The traditional EnKF with perturbed observations (Evensen
1994) introduces sampling errors by perturbing the observations. In contrast to the
traditional EnKF, the EnSRF (Whitaker and Hamill, 2002) and the Ensemble
Adjustment Kalman Filter (EAKF, developed by Anderson, 2001) obviate the need to
perturb the observations. The local ensemble Kalman filtering (LEKF), a kind of
EnSRF, was presented by Ott et al. (2002, 2004). It was computationally more
efficient compared to the traditional EnKF, since it simultaneously assimilates the
observations within a spatially local volume independently. The local Ensemble
Transform Kalman Filter (LETKF, Hunt, 2007) integrates the advantages of the
Ensemble Transform Kalman Filter (ETKF, developed by Bishop et al., 2001) and the
LEKF. The computational cost of LETKF is much lower than that of the original
LEKF because the former does not require an orthogonal basis. Though LETKF has
more advantages, we still chose the same EnSRF as Schwartz et al. (2014) because we
did not need to extend it to analyzing aerosol ICs, very similar to Schwartz et al.

(2014).

Following the notation of Ide et al. (1997), given an $m$-dimensional background
model forecast vector $\mathbf{x}^b$, a $p$-dimensional observation vector $\mathbf{y}^o$ and an operator $\mathbf{H}$
that converts the model state to the observation states, we expressed the variables as
an ensemble mean (denoted by an over-bar) and a deviation from the mean (denoted
by a prime). Thus, the ensemble mean $\bar{\mathbf{x}}^a$ of the analyzed state $\mathbf{x}^a$ and the
deviations $\mathbf{x}'^a$ from the ensemble mean are updated separately by
$$\bar{\mathbf{x}}^a = \bar{\mathbf{x}}^b + \mathbf{K}(\mathbf{y}^o - \mathbf{H}\bar{\mathbf{x}}^b), \text{(7)}$$
$$\mathbf{x}'^a = \mathbf{x}'^b + \widetilde{\mathbf{K}}(\mathbf{y}'^o - \mathbf{H}\mathbf{x}'^b), \text{(8)}$$
where $\mathbf{K}$ is the traditional Kalman gain matrix and $\widetilde{\mathbf{K}}$ is the gain used to update the
deviations from the ensemble mean. These are given by
$$\mathbf{K} = \mathbf{P}^b\mathbf{H}^T(\mathbf{H}\mathbf{P}^b\mathbf{H}^T + \mathbf{R})^{-1}, \text{(9)}$$

$$\widetilde{\mathbf{K}} = \mathbf{P}^b\mathbf{H}^T\left[\left(\sqrt{\mathbf{H}\mathbf{P}^b\mathbf{H}^T + \mathbf{R}}\right)^{-1}\right]^T \left(\sqrt{\mathbf{H}\mathbf{P}^b\mathbf{H}^T + \mathbf{R}} + \sqrt{\mathbf{R}}\right)^{-1}$$

$$= \left(1 + \sqrt{\mathbf{R}/(\mathbf{H}\mathbf{P}^b\mathbf{H}^T + \mathbf{R})}\right)^{-1} \mathbf{K}, \text{(10)}$$
where $\mathbf{P}^b = \frac{1}{N-1}\sum_{i=1}^{N}\mathbf{x}'^b(\mathbf{x}'^b)^T$ is the $m*m$-dimensional background error
covariance matrix and $\mathbf{R}$ is the $p*p$-dimensional diagonal observation error
covariance matrix. In real applications, $\mathbf{P}^b\mathbf{H}^T$ and $\mathbf{H}\mathbf{P}^b\mathbf{H}^T$ will be approximated
using the background ensemble; namely,
$$\mathbf{P}^b\mathbf{H}^T = \frac{1}{N-1}\sum_{i=1}^{N}\mathbf{x}'^b(\mathbf{H}\mathbf{x}'^b)^T \text{ (11)}$$
$$\mathbf{H}\mathbf{P}^b\mathbf{H}^T = \frac{1}{N-1}\sum_{i=1}^{N}\mathbf{H}\mathbf{x}'^b(\mathbf{H}\mathbf{x}'^b)^T. \text{(12)}$$
In equations (11) and (12), $N$ is the ensemble size.
Note that for the joint analysis of ICs and emissions, the state vector $\mathbf{x}$ is the
joint vector of the mass concentration $\mathbf{C}$ and the emission scaling factor $\boldsymbol{\lambda}$, i.e.
$\mathbf{x} = [\mathbf{C}, \boldsymbol{\lambda}]^T$. In this study, the state variables of the analysis of the ICs were the 15
WRF-Chem/GOCART aerosol variables, same as that reported by Schwartz et al.
(2012). The state variables of the emission scaling factors include $\boldsymbol{\lambda}_{PM2.5}$, $\boldsymbol{\lambda}_{SO2}$, $\boldsymbol{\lambda}_{NO}$
and $\boldsymbol{\lambda}_{NH3}$ and are described in section 2.3.1. After each ensemble analysis, the
ensemble forecasts were performed with the corresponding models to advance $\mathbf{C}$ and
$\boldsymbol{\lambda}$ to the next analysis time.
In this work, a 50-member ensemble was chosen, following Schwartz et al.
(2012) and Whitaker and Hamill (2002). Covariance localization forced EnSRF
analysis increments to zero 1280 km from an observation in the horizontal and one
scale height to reduce spurious correlations due to sampling error for all control
variables, similar to Pagowski et al., (2012) and Schwartz et al., (2012, 2014). In
addition, posterior (after assimilation) multiplicative inflation following Whitaker and
Hamill (2012) was applied aiming to maintain ensemble spread for only the
concentration analysis. The inflation factor $\alpha = 1.2$ was chosen as Pagowski et al.,
(2012) and Schwartz et al., (2012, 2014). Additive or prior inflation was not employed.
As for the emission scaling factor $\boldsymbol{\lambda}$, the inflation was not used at this step.

**2.3 Data assimilation system**
2.3.1 State variables

As stated in section. 2.2, the state variables of the analysis of the ICs were the 15

WRF-Chem/GOCART aerosol variables. The $PM_{2.5}$ observation operator was the
same as that described by Schwartz et al. (2012) and expressed as

$$\mathbf{y}^f = \boldsymbol{\rho}_d[\mathbf{P_{25}} + 1.375\mathbf{S} + 1.8(\mathbf{OC_1} + \mathbf{OC_2}) + \mathbf{BC_1} + \mathbf{BC_2}$$

$$+\mathbf{D_1} + 0.286\mathbf{D_2} + \mathbf{S_1} + 0.942\mathbf{S_2}], (13)$$
where $\boldsymbol{\rho}_d$ represents the dry air density, which is multiplied by the mixing ratios of
aerosol species (in $\mu g \cdot kg^{-1}$) to convert the units to $\mu g \ m^{-3}$ for consistency with the
observations.

From the perspective of the optimization of emissions, four species of emission

scaling factors ($\boldsymbol{\lambda}_{PM2.5}$, $\boldsymbol{\lambda}_{SO2}$, $\boldsymbol{\lambda}_{NO}$ and $\boldsymbol{\lambda}_{NH3}$) were also considered as the state
variables of the DA system. Atmospheric inorganic aerosols are not only from the
primary emissions, but also from secondary processes- chemical and thermodynamic
transformations from the gas-phase precursors. Therefore, not only the primary
sources of $PM_{2.5}$, but also the sources of the gas-phase precursors, need to be
optimized. In this study, the sources of $SO_2$, $NO_x$ and $NH_3$ ($\mathbf{E}_{SO2}$, $\mathbf{E}_{NO}$ and $\mathbf{E}_{NH3}$),
which have a large impact on the distribution of $PM_{2.5}$, were also optimized in
addition to the primary sources of $PM_{2.5}$. It is noted that for the optimization of the
emission scaling factors, $\mathbf{M}_{SF}$ serves as the forecast model and the observation
operator reflects the combined information of emissions (in the format of $\boldsymbol{\lambda}$ in
equation (6)), the physics and chemistry processes in WRF-Chem simulations and the
transformation $PM_{2.5}$ from model space to observation space (equation (13)).

The direct sources of $PM_{2.5}$ include the unspeciated primary sources of $PM_{2.5}$

$\mathbf{E}_{PM2.5}$, sulfate $\mathbf{E}_{SO4}$, nitrate $\mathbf{E}_{NO3}$, organic compounds $E_{org}$ and elemental
compounds $E_{BC}$; all of them are given in two modes (the nuclei and accumulation
modes, represented as i and j in the subscripts respectively). The ratios between the
nuclei and accumulation modes were the same as in the suggested emission process
for National Emission Inventory in WRF-Chem (Freitas et al., 2011). The formula of
sulfate and nitrate emissions in the model are as below:
$$\mathbf{E}_{PM2.5i} : \mathbf{E}_{PM2.5j} = 1 : 4, \ (14)$$

$$\mathbf{E}_{SO4i} : \mathbf{E}_{SO4j} = 1 : 4, \ (15)$$

$$\mathbf{E}_{NO3i} : \mathbf{E}_{NO3j} = 1 : 4, \ (16)$$

$$\mathbf{E}_{SO4i} + \mathbf{E}_{SO4j} = a * (\mathbf{E}_{PM2.5i} + \mathbf{E}_{PM2.5j} - \mathbf{E}_{EC} - \mathbf{E}_{ORG}), \ (17)$$

$$\mathbf{E}_{NO3i} + \mathbf{E}_{NO3j} = b * (\mathbf{E}_{PM2.5i} + \mathbf{E}_{PM2.5j} - \mathbf{E}_{EC} - \mathbf{E}_{ORG}), \ (18)$$

where $\mathbf{E}_{EC}$ represents elemental carbon and $\mathbf{E}_{ORG}$ organic compounds, and
$a = 0.074$ and $b = 0.038$ were chosen based on the internal emissions and
observational data. In the DA process, the first 6 species of direct sources of
emissions ($\mathbf{E}_{PM2.5i}$, $\mathbf{E}_{PM2.5j}$, $\mathbf{E}_{SO4i}$, $\mathbf{E}_{SO4j}$, $\mathbf{E}_{NO3i}$, and $\mathbf{E}_{NO3j}$), which may have
larger uncertainties in heavy polluted events, were updated according to the variation
of $\boldsymbol{\lambda}_{PM2.5}$. $\mathbf{E}_{PM2.5i}$ and $\mathbf{E}_{PM2.5j}$ were directly updated according to the variation in
$\boldsymbol{\lambda}_{PM2.5}$. The emissions ($\mathbf{E}_{SO4i}$, $\mathbf{E}_{SO4j}$, $\mathbf{E}_{NO3i}$ and $\mathbf{E}_{NO3j}$) were also updated according
to the variations in $\mathbf{E}_{PM2.5i}$ and $\mathbf{E}_{PM2.5j}$.

$\mathbf{E}_{EC}$ and $\mathbf{E}_{ORG}$ of the anthropogenic emissions were not assimilated, which is a

limitation in this work. Besides, emissions of dust and sea salt were not assimilated. It
is true that these emissions are also important for the atmosphere aerosol. The reason
we did not assimilate $\mathbf{E}_{EC}$ and $\mathbf{E}_{ORG}$ is that only the $PM_{2.5}$ measurements are used in
this DA experiment. However, the sources of the aerosols (especially organic aerosols)
are so complex that our knowledge of their formation mechanisms is far from clear.
Though it is technically possible to have all emissions assimilated, with such limited
observations adding more control variables would cause much more uncertainties in
the system which might lead to unreasonable analysis.

2.3.2 Procedure for the DA system
Figure 2 (b) shows the workflow of the DA system. The steps in this workflow are as
follows.
(1) The persistence forecasting operator $\mathbf{M}_{SF}$ is applied to forecast the
background fields of the emission scaling factors $\boldsymbol{\lambda}_{PM2.5}^{f}$, $\boldsymbol{\lambda}_{SO2}^{f}$, $\boldsymbol{\lambda}_{NO}^{f}$ and $\boldsymbol{\lambda}_{NH3}^{f}$. The
forecast chemical fields of $P_{25}$, $SO_2$, NO and $NH_3$ of the previous assimilation cycle
are used to create the prior emission scaling factors $\boldsymbol{\lambda}_{PM2.5}^{p}$, $\boldsymbol{\lambda}_{SO2}^{p}$, $\boldsymbol{\lambda}_{NO}^{p}$ and $\boldsymbol{\lambda}_{NH3}^{p}$.
The background scaling factors are then generated using equation (5).
(2) The ensemble members of the emissions, $\mathbf{E}_{PM2.5i}^{f}$, $\mathbf{E}_{PM2.5j}^{f}$, $\mathbf{E}_{SO2}^{f}$, $\mathbf{E}_{NO}^{f}$ and
$\mathbf{E}_{NH3}^{f}$, are prepared according to equation (6). The corresponding emissions of $\mathbf{E}_{SO4i}^{f}$,
$\mathbf{E}_{SO4j}^{f}$, $\mathbf{E}_{NO3i}^{f}$ and $\mathbf{E}_{NO3j}^{f}$ are obtained based on equations (15–18). Other inorganic
species of the anthropogenic emission, such as $\mathbf{E}_{EC}$ and $\mathbf{E}_{ORG}$, are not perturbed for
WRF-Chem. However, other anthropogenic emissions, such as $\mathbf{E}_{PM2.5}$, $\mathbf{E}_{SO4}$ and
$\mathbf{E}_{NO3}$, are much larger than $\mathbf{E}_{EC}$ and $\mathbf{E}_{ORG}$ in most area of China, and the ensemble
spreads of the aerosol concentrate largely dependent on the uncertainties of those
anthropogenic emissions. Besides, model errors raised from the meteorology, the
emission and the chemical model itself are compensated to some extent through the
use of multiplicative inflation. In other words, the ensemble spread of the
concentrations can be kept at a certain level though $\mathbf{E}_{EC}$ and $\mathbf{E}_{ORG}$, are not
perturbed.
Natural emissions, such as dust and sea salt emissions were not perturbed
explicitly when the forecast emissions were generated. However, emissions of dust
and sea salt were parameterized within the GOCART model (Chin et al., 2002).
Within the DA system, varying meteorology across the members implicitly perturbed
dust and sea salt emissions.

(3) Forced by the changed emissions ($\mathbf{E}_{PM2.5i}$, $\mathbf{E}_{PM2.5j}$, $\mathbf{E}_{SO2}$, $\mathbf{E}_{NO}$, $\mathbf{E}_{NH3}$,

$\mathbf{E}_{SO4i}$, $\mathbf{E}_{SO4j}$, $\mathbf{E}_{NO3i}$ and $\mathbf{E}_{NO3j}$ were substituted by $\mathbf{E}^f_{PM2.5i}$, $\mathbf{E}^f_{PM2.5j}$, $\mathbf{E}^f_{SO2}$, $\mathbf{E}^f_{NO}$,
$\mathbf{E}^f_{NH3}$, $\mathbf{E}^f_{SO4i}$, $\mathbf{E}^f_{SO4j}$, $\mathbf{E}^f_{NO3i}$ and $\mathbf{E}^f_{NO3j}$; the other emissions such as $\mathbf{E}_{EC}$ and $\mathbf{E}_{ORG}$
remained unchanged), WRF-Chem is run again to forecast the chemical fields $\boldsymbol{\rho}^f$
with the updated chemical fields of the previous assimilation cycle as the ICs. The
state variables, i.e., 15 aerosol species and four scaling factors, are then prepared.

(4) The model-simulated PM$_{2.5}$ concentration at the observation space is then

calculated via equation (13). At this time, the state vector $\mathbf{x}^f = [\mathbf{C}^f, \boldsymbol{\lambda}^f]^T$ was
prepared.

(5) In the assimilation step, the state variables, the concentrations of 14 defined

aerosol species and a 15th unspeciated aerosol, and the four species of emission
scaling factors $\boldsymbol{\lambda}^f_{PM2.5}$, $\boldsymbol{\lambda}^f_{SO2}$, $\boldsymbol{\lambda}^f_{NO}$ and $\boldsymbol{\lambda}^f_{NH3}$, were optimized through EnSRF.

(6) After the assimilation step, the optimized emissions ($\mathbf{E}^a_{PM2.5i}$, $\mathbf{E}^a_{PM2.5j}$, $\mathbf{E}^a_{SO2}$,

$\mathbf{E}^a_{NO}$, $\mathbf{E}^a_{NH3}$, $\mathbf{E}^a_{SO4i}$, $\mathbf{E}^a_{SO4j}$, $\mathbf{E}^a_{NO3i}$ and $\mathbf{E}^a_{NO3j}$) were calculated according to equations
(6, 15–18) using the optimized scaling factors ($\boldsymbol{\lambda}^a_{PM2.5}$, $\boldsymbol{\lambda}^a_{SO2}$, $\boldsymbol{\lambda}^a_{NO}$ and $\boldsymbol{\lambda}^a_{NH3}$).

**3. PM$_{2.5}$ observation data and errors**
Hourly averaged surface PM$_{2.5}$ observations from the Ministry of Environmental
Protection of China were assimilated. There were altogether 876 national control
measurement sites over China. The PM$_{2.5}$ observation sites spanned most of central
and eastern China but were primarily located in urban and suburban areas. So it
always happened that there were more than one observation sites in certain city,
which were fall into the same model grid. Since we did not know the exact
observation environment of the sites, we randomly selected one observation site in a
city for assimilation experiment and one for verification purposes to ensure that there
was at most one assimilated measurements for one model grid. Altogether 77 stations
were selected for the PM$_{2.5}$ assimilation experiment and another 77 independent
stations were selected for verification. Figure 1 shows the locations of 77
measurement sites used for the PM$_{2.5}$ assimilation experiment and 77 independent
sites used for forecast verification.

The observation error covariance matrix **R** in equation (9) includes

contributions from measurement and representation errors. Similar to the work of
Schwartz et al. (2012), the measurement error $\varepsilon_0$ is defined as $\varepsilon_0 = 1.5 + 0.0075 *$
$\Pi_0$, where $\Pi_0$ denotes the observational values for PM$_{2.5}$ (µg m$^{-3}$). Thus, higher
PM$_{2.5}$ values were associated with larger measurement errors. Following Elbern et al.
(2007) and Pagowski et al. (2010), Schwartz et al. (2012), the representativeness error
$\varepsilon_r$ depends on the resolution of the model and the characteristics of the observation
locations and is calculated as $\varepsilon_r = r\varepsilon_0\sqrt{\Delta x/L}$, where $r$ is an adjustable parameter
(here, $r = 0.5$), $\Delta x$ is the grid spacing (here, 40.5 km), and L is the radius of
influence of an observation (here, L was set to 3 km following Elbern et al. (2007),
since we do not know the station type that used in this work). The total PM$_{2.5}$ error ($\varepsilon_t$)
is defined as $\varepsilon_t = \sqrt{\varepsilon_0{}^2 + \varepsilon_r{}^2}$. The observation errors are assumed to be uncorrelated
so that **R** is a diagonal matrix.

The PM$_{2.5}$ observations were subject to quality control to ensure data reliability

before DA. Considering that China has had intense pollution events, PM$_{2.5}$ values
larger than 800 µg m$^{-3}$ were classified as unrealistic and were not assimilated;
observations with the ensemble mean of the first guess departure exceeding 100
µg m$^{-3}$ were also omitted, following Schwartz et al. (2012). The numbers of the
observations were about 17700. Among them 8 observations were discarded because
they were larger than 800 µg m$^{-3}$ and 243 (around 1.5%) were discarded due to the
latter reasons.

**4. Experimental design**
Two parallel experiments were performed to evaluate the impact of PM$_{2.5}$ DA on the
analyses and forecasts of aerosols over China: an assimilation experiment and a
control experiment. Both experiments used identical WRF-Chem settings and
physical parameterizations.

4.1 Spin-up ensemble forecast with perturbed Initial and boundary conditions
The initialization and spin-up procedures were identical to those reported by
Schwartz et al. (2014). The ICs and lateral boundary conditions (LBCs) for the
meteorological fields were provided by the National Centers for Environmental
Prediction Global Forecast System (GFS).
The initial meteorological fields were created at 0000 UTC 1 October 2014 by
interpolating the GFS analyses onto the model domain. The 50 ensemble members
were then generated by adding Gaussian random noise with a zero mean and static
background error covariances (Torn et al., 2006) to the temperature, water vapor,
velocity, geopotential height and dry surface pressure fields. The ICs of each member
were zero in the initial aerosol fields, representing clean conditions as described by
Liu et al. (2011).
The LBCs for the meteorological fields were then interpolated from the GFS
analyses from 0000 UTC 1 October 2014 to 0000 UTC 16 October 2014 and
perturbed similarly to the initial fields at 0000 UTC 1 October 2014. The aerosol
LBCs of each member for all experiments were idealized profiles embedded within
the WRF/Chem model.
Fifty-member emissions were created by adding random noise to the
anthropogenic emissions, same as reported by Schwartz et al. (2014),

$$\mathbf{E}_{ip}^*(\eta, t) = \mathbf{E}_p(\eta, t) + \mathbf{W}_{ip}\boldsymbol{\sigma}_p^{\mathrm{E}}(\eta, t)$$

where $\mathbf{E}_{ip}^*(\eta, t)$ is the $i$th ensemble member for the $p$th emissions variable at the
$\eta$th grid point and the $t$th hour, $\mathbf{E}_p$ is the unperturbed emissions. The term $\boldsymbol{\sigma}_p^{\mathrm{E}}$ is
the standard deviation of all $\mathbf{E}_p$ values and in the horizontally adjacent points of grid
box $\eta$ at and within 2 h of $t$. $\boldsymbol{W}$ is a weight that was randomly drawn from a
standard Gaussian distribution and varied for each ensemble member and variable but
was spatially and temporally constant. No correlations between emissions variables
were considered, which was a limitation of this approach. For possible negative
perturbed emissions, they were set as $\mathbf{E}_{ip}^{*}(\eta, t) = 0.001 * \mathbf{E}_p(\eta, t)$. This will increase
the prescribed emissions more or less. However, only very few data were negative. So,
this influence can be negligible.

Before the first DA cycle, a 50-member ensemble of four-day WRF-Chem

forecasts was performed from 0000 UTC 1 October to 2300 UTC 4 October 2014
using the perturbed ICs at 0000 UTC 1 October 2014, the corresponding perturbed
LBCs and the emissions. Then a 50-member ensemble aerosol forecasts at 0000 UTC
5 October 2014 were produced.

4.2 Assimilation experiments
Two DA experiments were performed. One was the pure assimilation of chemical ICs
(hereafter expC), the others was the joint adjustment of chemical ICs and source
emissions (hereafter expJ). Both DA experiments had same settings except for the
emissions. They were conducted from 0000 UTC 5 October 2014 to 0000 UTC 16
October 2014. The assimilation cycle interval was 1 h.

In the first DA cycle in expJ, the first 50 ensemble chemical fields were drawn

from the WRF-Chem ensemble forecasts valid at 0000 UTC 5 October 2014, as
described in section 4.1. Using the ensemble aerosol forecasts, the prior emission
scaling factors $\boldsymbol{\lambda}_{i,t}^{\mathrm{p}}$ at 2300 UTC 4 October 2014 were calculated. $\boldsymbol{\lambda}_{i,t}^{\mathrm{p}}$ were used
directly as $\boldsymbol{\lambda}_{i,t}^{\mathrm{f}}$ for the first 5 assimilation cycles (after 5 assimilation cycles, the
system has been initialized, all future scaling factors could be created using the
persistence forecasting operator $\mathbf{M}_{\mathrm{SF}}$). Then, the state vector $\mathbf{x}^{\mathrm{f}} = [\mathbf{C}^{\mathrm{f}}, \boldsymbol{\lambda}^{\mathrm{f}}]^{\mathrm{T}}$ was
prepared. And after that, the DA cycle started.

In expC, the first chemical fields were also drawn from the WRF-Chem

ensemble forecasts valid at 0000 UTC 5 October 2014. Then, the state vector
$\mathbf{x}^{\mathrm{f}} = [\mathbf{C}^{\mathrm{f}}]^{\mathrm{T}}$ was prepared and the DA cycle started.

At the WRF-Chem forecast step of the subsequent assimilation cycles for both

experiments, the ICs for the chemical variables of each member were drawn from the
updated chemical fields of the previous cycle. The aerosol LBCs of each member for
all experiments were idealized profiles embedded within the WRF/Chem model. As
for the meteorological ensemble fields, the LBCs were prepared in advance as
depicted in section 4.1; the ICs of each member of the meteorological fields were
drawn from the forecast meteorological fields of the previous cycle before
re-centering with the GFS analysis because we do not do meteorological analysis:
$$\boldsymbol{\pi}_{i_{\text{new}}} = \boldsymbol{\pi}_i + (\boldsymbol{\pi}_{\text{GFS}} - \overline{\boldsymbol{\pi}}), \text{ (18)}$$
where $\boldsymbol{\pi}_i$ is the $i$th member of the forecast meteorological fields of the previous
cycle, $\overline{\boldsymbol{\pi}}$ is the ensemble mean of the forecast meteorological fields of the previous
cycle, $\boldsymbol{\pi}_{\text{GFS}}$ is the meteorological field interpolated from the GFS analyses and
$\boldsymbol{\pi}_{i_{\text{new}}}$ is the new meteorological field used as the IC in WRF-Chem in the next cycle.

As stated in the first paragraph in this section, the settings of expC were the same

as those in expJ except for the emissions. In expJ, the ensemble anthropogenic
emissions were generated by using emission scaling factors. While in expC, the
ensemble anthropogenic emissions were prepared by adding random noise, as stated
in 4.1.

4.3 Control experiment
The control experiment was conducted for the same period as the assimilation
experiment and the simulation cycle period was 1 h, as in the assimilation experiment.
The first initial chemical fields were extracted from the ensemble mean valid at 0000
UTC 5 October 2014. In the subsequent simulation process, the ICs for the chemical
fields were from the previous cycle's 1-h forecast. The LBCs and ICs for the
meteorological fields were updated by interpolating the GFS analyses. The emissions
were the prescribed emissions $\mathbf{E}_t^{\text{p}}$ without any perturbation.

**5. Results**
Statistics for both expJ and expC were computed using the ensemble mean prior
(background) and posterior (analysis) fields (average of the 50-member ensemble).
The ensemble performances were first examined. Output from the first day of the
cycling DA configurations was excluded from all verification statistics to allow the
ensemble fields to ''spin up'' from the initial ensemble.
As the measurement coverage is an important factor that may determine the
performance in DA, we primarily focused our attention on the results from three
sub-regions with comparatively dense observational coverage (Figure 1): the Beijing–
Tianjin–Hebei region (JJJ, 12 stations for assimilation and 12 stations for verification);
the Yangtze River delta (YRD, 24 stations for assimilation and 24 stations for
verification); and the Pearl River delta (PRD, 9 stations for assimilation and 9 stations
for verification).

5.1 Ensemble performance
It is important to assess the ensemble performance for an ensemble-based DA system.
In a well-calibrated system, a comparison of the prior ensemble mean
root-mean-square error (RMSE) with respect to the observations should equal the
prior "total spread" (square root of the sum of ensemble variance and observation
error variance) (Houtekamer et al., 2005). Figure 3 shows the time series for the prior
ensemble mean RMSE and the total spread for $PM_{2.5}$ aggregated over all observations
in the three sub-regions for expJ. It indicates that the magnitudes of both the total
spread and the RMSE were influenced by the diurnal cycle and heavy air pollution.
Almost all the total spreads were smaller than the RMSE, showing an insufficient
spread of $PM_{2.5}$ ensemble forecasts, which is especially evident for heavy polluted
period with much larger RMSEs. For expC, the characteristics of the prior ensemble
mean RMSE and the total spread for $PM_{2.5}$ were very similar to that for the joint DA
experiment.
The magnitudes of the ensemble spread of the emission scaling factors of the
joint DA experiment were important for emission inversion. They were very stable
throughout the ~10 day experiment period, which indicates that $\mathbf{M}_{SF}$ can generate
stable artificial data to generate the ensemble emissions. For $\boldsymbol{\lambda}^{f}_{PM2.5}$, they ranged
from 0.25 to 1 in most model area. Figure 3d shows the area-averaged time series
extracted from the ensemble spread of $\lambda_{PM2.5}^f$. It shows that the ensemble spread was
stably distributed around 0.5, which indicates that the uncertainty of the ensemble
emissions was about 50%.

5.2 Impact on aerosol ICs
To evaluate quantitatively the impact of the ensemble assimilation system on the ICs,
the mean errors (bias), RMSEs and correlation coefficient (CORR) of the assimilation
experiment and the control run were first analyzed. These statistics were calculated
against independent observations over all the analyses from 6 to 16 October 2014.
Table 1 shows that the bias magnitudes of the control run were 15.9 and 20.6 μg m$^{-3}$
for the YRD and the PRD, respectively, suggesting a significant overestimation of the
WRF-Chem aerosol mass in these two sub-regions. However, a significant
underestimation of the aerosol mass occurred in the JJJ region, where the model bias
was −18.0 μg m$^{-3}$. The RMSEs of the control run were 81.6, 30.6 and 31.8 μg m$^{-3}$ for
the JJJ, YRD and PRD regions, respectively. After assimilation, the statistics showed
an apparent improvement and the magnitude of the bias and the RMSE decreased for
both DA experiment. For expJ, both the maximum bias and the RMSE were obtained
in the JJJ region, and were -10.3 and 66.9 μg m$^{-3}$, respectively. The CORR increased
from 0.79, 0.60, and 0.62 to 0.83, 0.85, and 0.80 for the JJJ, YRD and PRD,
respectively. The statistics of expC were very similar to those of expJ. The bias and
the RMSE in the JJJ region were -12.2 and 64.0 μg m$^{-3}$, respectively. And the CORR
were 0.85, 0.80, and 0.80 for the JJJ, YRD and PRD, respectively. These results
indicate that the initial PM$_{2.5}$ fields can be adjusted efficiently by the EnSRF.
It is interesting to note that expC has better RMSE and CORR than expJ but poor
bias in JJJ. And expC has better bias and RMSE than expJ but poor CORR in PRD.
Maybe small number of samples caused the uncertainties of the statics. However, the
differences were very small. The analysis of both experiments were very similar.
Then the analysis increments (i.e. $\bar{\mathbf{x}}^a - \bar{\mathbf{x}}^b$) were investigated to show the direct
impact of PM$_{2.5}$ DA. They are determined by both the observation increments and the
relative magnitudes of the forecast error and the observation error, based on Equation
(7). From Figure 4(a), (e) and (f), the increments of both assimilation experiments
were distributed around the observations as expected. However, the impact of
assimilating $PM_{2.5}$ observations was not limited to the areas where observations were
located, observations information was also transported to other areas through the
WRF-Chem forecast. Besides, the ensemble forecasts also partly contributed to the
spatial distribution of the $PM_{2.5}$ mass. Therefore, the spatial distributions of the $PM_{2.5}$
mass in both assimilation experiments were significantly different from the control
run (see Figure 4(b), (c) and(d)), which suggest that assimilation $PM_{2.5}$ observations
impacts greatly on the aerosol ICs. The $PM_{2.5}$ mass magnitude of both assimilation
experiments were smaller than that of the control run at the lowest model level in the
YRD, the PRD and in central China. Conversely, positive differences (analysis minus
control) were gained in the JJJ region and in northeast China. These indicated the
reduction of the overestimation or underestimation of the WRF-Chem simulation over
these regions with data assimilation.

5.3 Impact on emissions
To determine the impact of assimilating $PM_{2.5}$ observations on the chemical emissions,
we analyzed the area-averaged time series extracted from the forecast emission
scaling factors, the optimized emission scaling factors, the prior emissions and the
optimized emissions. Figure 5 shows that $\boldsymbol{\lambda}^{f}_{PM2.5}$ were changed along with $\boldsymbol{\lambda}^{a}_{PM2.5}$.
This indicates that observation information ingested from the previous observations
was incorporated through the usage of the time smooth operator.

Figure 5 also shows that although the prior emissions $\mathbf{E}^{p}_{PM2.5}$ had no diurnal

variation when the experiments were designed, the optimized $PM_{2.5}$ scaling factor,
$\boldsymbol{\lambda}^{a}_{PM2.5}$, showed an obvious variation with time, as did the optimized unspeciated
primary sources of $PM_{2.5}$, $\mathbf{E}^{a}_{PM2.5}$. Moreover, the values of $\boldsymbol{\lambda}^{a}_{PM2.5}$ were <1 at almost
all times in the YRD and PRD, which resulted that the analyzed emission $\mathbf{E}^{a}_{PM2.5}$
were lower than the prior $PM_{2.5}$ emissions $\mathbf{E}^{p}_{PM2.5}$. In the YRD, the prior $\mathbf{E}^{p}_{PM2.5}$ was
about 0.127 μg m$^{-2}$ s$^{-1}$ over all hours. After assimilation, the time-averaged optimized
$\mathbf{E}_{PM2.5}^a$ decreased to 0.107 μg·m$^{-2}$s$^{-1}$, about 15.6% lower than the prior value. In the
PRD, the prior $\mathbf{E}_{PM2.5}^p$ was about 0.10 μg m$^{-2}$ s$^{-1}$. The time-averaged optimized
$\mathbf{E}_{PM2.5}^a$ decreased to 0.066 μg·m$^{-2}$ s$^{-1}$, leading to a decrease of 35.0%. However,
larger values for the optimized $\mathbf{E}_{PM2.5}^a$ were obtained in the JJJ region in three
periods, from 1600 UTC 6 October to 0000 UTC 8 October, from 1600 UTC 9
October to 0000 UTC 10 October, and from 1600 UTC 13 October to 0000 UTC 15
October as a result of the increased optimized scaling factor $\boldsymbol{\lambda}_{PM2.5}^a$. This may have
been caused by the burning of crop residues during harvesting in this region (Li et al.,
2016), which was not taken into account in the prior emissions. However, the PM$_{2.5}$
measurements network was still spatially sparse and heterogeneous in this work.
Almost all measurements were located in the city and no data available in the rural.
Meanwhile, the crop residues burning always occur in the rural region. Therefore, the
PM$_{2.5}$ measurements network can only capture the burning information a few hours
later. Hence, although the system is able to detect the emission changes caused by
burning events, the time that the system started to show increased scaling factors
might be not accurate enough (may shift a few hours later). Maybe a Kalman
smoother would have been a better system to solve this problem.
The NO, SO$_2$ and NH$_3$ emissions were all adjusted to some extent by our DA
approach (see Figure 6). The NO emissions increased by 41.3, 43.7 and 20.3% in the
JJJ, YRD and PRD regions, respectively. The SO$_2$ emissions increased by 16.3, 10.0
and 18.3% and the NH$_3$ emissions increased by 16.7, 7.8 and 7.5% in the JJJ, YRD
and PRD regions, respectively.
Figure 7 shows the spatial distribution of the time-averaged scaling factors
$\boldsymbol{\lambda}_{PM2.5}^a$ at the lowest model level over all hours from 6 to 16 October 2014, since the
emissions at higher levels were so small that the impact of assimilating PM$_{2.5}$
observations was negligible. Figure 8 shows the distribution of $\mathbf{E}_{PM2.5}^p$ and the
time-averaged differences between the ensemble mean of the assimilation and the
prior values.
These patterns are consistent with those in Figure 5. Negative differences were
obtained in most areas of the YRD and PRD, indicating that the $PM_{2.5}$ DA primarily
decreased the $PM_{2.5}$ emissions. Conversely, positive differences were obtained in
South Hebei, North Henan and Southeast Shanxi provinces, indicating that DA
increased the $PM_{2.5}$ emissions.

As the economy in China has developed, the spatiotemporal distribution of

emissions has changed as a result of changes in energy consumption, the structure of
the energy market and advances in technology. Therefore although this inventory of
emissions may have correctly described anthropogenic emissions in 2006 when it was
constructed, it is not representative of the anthropogenic emissions in 2014.
Theoretically, the assimilated emissions should reduce the uncertainty in the prior
emissions as a result of the application of observations. Different from the situations
that standard national emission inventories were reported by government in USA,
European or other countries, the rapid economic development and complexity of
emission sources in China lead to large uncertainties in the current emission
inventories even for the latest version. Thus it's impossible for us to conduct the direct
evaluation on emissions.

Although we had no direct emission observations to evaluate the analyzing

emissions, which was a challenging to many emission inversion research teams (e.g.
Tang et al, 2011; Miyazaki et al., 2012; Ding et al., 2015; Mclinden et al., 2016; etc.),
the improvement of emissions can be verified in terms of two aspect, the diurnal
variation and the location of increased emissions. The diurnal variation in the
assimilated emissions verified this statement to some extent. Especially in the PRD
and YRD, $\mathbf{E}^{a}_{PM2.5}$ in the daytime were always larger than those in the night, which
agreed well with Olivier et al. (2003), the WRAP (2006) and Wang et al. (2010). In
addition, the locations of the larger values for the optimized $E^{a}_{PM2.5}$ in the JJJ region
was in good agreement with the place of the crop residues burning *traced by the*
*environmental satellite of China.* There were 10, 231, 37 and 3
crop residue burning spots in Hebei, Henan, Shandong and Shanxi province
respectively from 5 to 11 October 2014 and 7, 20, 5 and 21 respectively from 12 to 18
October 2014 (Weekly Crop Residue Burning Monitoring Report traced by
Environmental Satellite, 2015a, 2015b).

However, the analysis emissions are only a mathematical optimum. They are

influenced greatly by the model errors and the observation errors. In addition, only
surface $PM_{2.5}$ observations were applied in this work, which may lack abundant
constraint on the sources of the secondary aerosol precursors. More observations are
needed to obtain reliable emissions for the sources of the gas-phase precursors.

5.4 Verification of aerosol forecasting

For the assimilation experiment, 48-h forecasts were performed at each 0000

UTC from 6 to 16 October 2014 with the hourly forecast output for both assimilation
experiments. For the verification forecasting experiment for expJ (hereafter fcJ), the
ensemble mean of the analyzed ICs and emissions of expJ were used in this
longer-range model forecast. For the verification forecasting experiment for expC
(hereafter fcC), the ensemble mean of the analyzed ICs of expC and the prescribed
anthropogenic emissions were used.

In order to get a more visualized picture of the impact of DA for both

assimilation experiments, time series of the hourly $PM_{2.5}$ extracted from the analysis
(AN), the control run (CT) and the hourly output of 48-h forecast (fc24 for the first
day forecast and fc48 for the second day forecast) were compared with the
observations (OBS) for three megacities Beijing, Shanghai and Guangzhou,
respectively (Figure 9). As expected, the time series of the analysis (also the
background) were consistent with the observations. The control run showed large
deviations from the observations, especially in Shanghai and Guangzhou. Benefit
from DA on both the first day and the second day forecasts can be clearly seen.

The bias and the RMSE of the surface $PM_{2.5}$ forecasts as a function of forecast

range was then calculated against the independent observations for the three
sub-regions (Figure 10). Both the bias and the RMSEs of the control run were
characterized by the diurnal cycle in the YRD and PRD. The largest errors were seen
at 2100 UTC in the YRD (about 29 $\mu g \cdot m^{-3}$ for bias and 37 $\mu g \cdot m^{-3}$ for RMSEs) and at
2300 UTC in the PRD (about 36 $\mu g \cdot m^{-3}$ for bias and 41 $\mu g \cdot m^{-3}$ for RMSEs), likely
indicating significant systematic forecast errors at these times. From 0300 to 0900
UTC, the bias (about 1 μg·m$^{-3}$ in the YRD and -5 μg·m$^{-3}$ in the PRD) and the RMSE
values (about 14 μg·m$^{-3}$ in the YRD and 16 μg·m$^{-3}$ in the PRD) were much smaller
than at other times in both the YRD and PRD, showing that WRF-Chem performed
well during this period. However, in the JJJ region, the bias (about -20 μg·m$^{-3}$) and
the RMSEs (about 50 μg·m$^{-3}$) were always large as a result of a heavy pollution event.
After assimilation, both the magnitude of the bias and the RMSEs decreased sharply.
Especially in in YRD and PRD, most bias ranged from -5 to 5 μg·m$^{-3}$ and most
RMSEs ranged from 11 to 14 μg·m$^{-3}$, further indicating that DA greatly affected the
ICs.

The improvements in the surface PM$_{2.5}$ forecasts by the joint adjustment of the

ICs and emissions were very large in the YRD and PRD for expJ. Large reduction of
the magnitude of the bias and the RMSEs due to assimilation can be seen for almost
the entire 48-h forecast range. From 10- to 23-h and from 34- to 47-h, in particular,
the relative reduction in RMSE was about 37.5%. However, the DA impact was much
smaller for 3- to 9-h forecast ranges, which are at daytime of the first day forecast. In
addition, the improvements were nearly negligible in PRD from 27- to 33-h, the
daytime of the second day forecast, suggesting that the benefit gained from adjusting
the ICs decreased progressively and eventually disappeared with model integration.
And the performance was actually deteriorated in YRD during the same time. One of
the possible reasons was that chemical model performed sufficiently well during
daytime when the boundary layer was unstable and therefore the further improvement
was more difficult. And there were always large errors during the night when the
boundary layer was stable, so that large improvements could be obtained. The other
possible reason can be attributed to the a priori constant emissions. The differences
between the optimized PM$_{2.5}$ emissions and the prior emissions were comparatively
small during the day, but the optimized PM$_{2.5}$ emissions were much smaller than the a
prior emissions during the night. So that the control run could performed worse during
the night and it could performed well during the day. Given the a priori variable
emissions provided, the control run will perform better during the night. Nevertheless,
attributed greatly to the large adjustment of chemical emissions, substantial
improvements were still achieved from 34- to 47- h. These results revealed that joint
adjustment of the ICs and emissions can improve surface PM$_{2.5}$ forecasts up to 48 h in
the YRD and PRD.
As for expC, it seemed that large improvements in the surface PM$_{2.5}$ forecasts
were gained through the adjustment of the ICs in PRD from 10- to 23-h and from 34-
to 47-h. Large reduction of the magnitude of the bias and the RMSEs due to
assimilation can be seen during this period. The relative reduction in RMSE ranged
from 25% to 37.5%. However, the forecasts deviated much from the observations for
3- to 9-h and 27- to 33-h forecast ranges. One of the reason may be that the
adjustment of the ICs decreased the analysis field too much on the whole since the
WRF-Chem forecast aerosol mass was systematically overestimated in PRD (see
Figure 4, Figure 9f and Figure 10e). While this aerosol mass overestimation might be
also due to the possibly overestimated emissions in some time periods (not all-day
long) which are not corrected in the simulation. So the over-adjusted ICs compensated
the unadjusted emissions in some period but also lead to the negative biases for the
periods when emission is not overestimated or underestimated. The other factor was
the diurnal variation. It is very clear that PM$_{2.5}$ mass gradually decreased with time
from 0000 UTC to 0008 UTC and then obtained the smallest value. After that it
increased with time from 0009 UTC to 0023 UTC obtained the largest value at about
0000 UTC. Both reasons led to the systematically underestimation of PM$_{2.5}$ mass of
fcC from 3- to 9-h and from 27- to 33-h, though maybe the aerosol ICs were very
close to the observations. Therefore, both the magnitude of the bias and the RMSEs of
the fcC were larger than those of the control run. In addition, PM$_{2.5}$ forecasts of the
fcC were benefit much from the diurnal variation and the adjustment of the ICs from
10- to 23-h and from 34- to 47-h. As a consequence, the magnitude of the
corresponding bias and the RMSEs of the fcC were smaller than those of the control
run. Similar statics characteristics were also gained in YRD. But the improvements
were comparatively small from 10- to 23-h and from 34- to 47-h. However, the
performance of fcJ was much better than that of fcC during the night in PRD and
YRD. While in the daytime, the improvement of expJ seems to be not so big or even
negligible. This could be attributed much to the emissions since the ICs of both
forecasts were very similar. In the forecast experiment of expC, the emissions were
the default monthly anthropogenic emissions. While in the forecast experiment of
expJ, the assimilated emissions were much smaller than the default anthropogenic
emissions in almost all the night in both regions indicating that the prior emission
uncertainties might be the dominating reasons that cause biases between observed and
model simulated concentrations in these cases. In the daytime in PRD, the assimilated
emissions were a little smaller than the default anthropogenic emissions. But in the
daytime in YRD, the assimilated emissions were a little larger than the default
anthropogenic emissions for most time (see Figure 5). While those changes between
assimilated emissions and prior emissions in the daytime are not as significant as that
in the nighttime.
Both DA systems did not perform as well in the JJJ region as in the YRD and
PRD. Relatively smaller improvements were achieved in the first 24-h forecast but
then no improvements were achieved afterwards in JJJ. One possible reason for this
result may be systematic errors due to chemistry mechanism in WRF-Chem. The
sources of the aerosols are so complex that our knowledge of their formation
mechanisms is far from clear and large uncertainties still exist in the model
simulations. Chemical transport models have a tendency to underestimate PM
concentrations, especially during episodes of heavy pollution (Denby et al., 2007) due
to some missing reactions (Wang et al., 2014; Zhang et al., 2015, Zheng et al., 2015;
Chen et al., 2016). Another reason can be attributed to the forecast meteorological
fields. There were still large uncertainties, especially when boundary layer was stable
and the wind speed was very small during episodes of heavy pollution. As a result, a
large bias may be obtained in forecasts of heavy pollution given the ICs and emission
inventories achieved from the joint assimilation. Another reason may be the sparse
coverage of measurements. There were only 12 sites in the JJJ region (Figure 1) and
the measurement coverage was much sparser than in the YRD or PRD.

**6. Summary and Discussion**

The EnSRF algorithm was extended to adjust the chemical ICs and the primary and precursor emissions to improve forecasts for surface $PM_{2.5}$. This system was applied to assimilate hourly surface $PM_{2.5}$ measurements from 5 to 16 October 2014 over China. To evaluate the effectiveness of DA, 48-h forecasts were performed using the optimized ICs and emissions, together with a control experiment without DA. Besides, the experiment of pure assimilation chemical ICs and the corresponding 48-h forecasts experiment were also performed for comparison. The results indicated that the forecasts with the optimized ICs and emissions performed much better than the control simulations. Large improvements were achieved for almost all the 48-h forecasts, particularly in the YRD and PRD. However, it did show some improvements in the first 24-h but then there is no difference between the control run and the forecasts in the JJJ region afterwards, which may be attributed to the sparse measurement coverage and the deficiencies in the model system for forecasting heavy pollution. Comparing to the forecasts with only the optimized ICs, the forecasts with the joint adjustment were always much better during the night in the PRD and YRD regions. However, they were very similar during daytime in both regions. And in the JJJ region, they performed similarly for almost the entire 48-h forecast range.

There are still some limitations in this study. Firstly, we use the default monthly anthropogenic emissions as the prior emissions and no time variation was added to keep objective, since no resolution of temporal allocations at shorter but critical (e.g.,day-of-week, diurnal) scales is available. As shown in earlier work, the constant emissions will worsen the chemical forecasts (de Meij et al., 2006; Wang et al, 2009). For the joint DA system itself, it cannot benefit from the constant prior anthropogenic emissions. But the normalized RMSE in Figure 10g decreased due to the poor forecasts of control run. The control run will perform better when variable emissions within the day are allowed, especially during the night. As a result, the relative reduction in RMSE could not be so large during the night. Secondly, no correlations between emissions variables were considered when perturbing the emissions, which

will lead to the reduction of the correlations between the variables. Thus, the chemical
forecast will deviate from the truth to some degree. Fortunately, the perturbed
emissions were only used in the initialization and spin-up experiment and expC.
Therefore, there were no impacts on expJ and the control run except for expC. Thirdly,
$E_{EC}$ and $E_{ORG}$ are not perturbed in expJ. However, as stated in Sect. 2.3.2, the
ensemble spread of $OC_1$ and $OC_2$ can be kept at a certain level. As a result, $OC_1$
and $OC_2$ changed much contributed to the $PM_{2.5}$ assimilation in expJ, which
suggests that the influence of not perturbing $E_{EC}$ and $E_{ORG}$ could be negligible. But,
because of the too small magnitudes of $BC_1$ and $BC_2$, the differences (assimilation
minus control) of $BC_1$ and $BC_2$ were nearly close to zero. Fourthly, the experiment
(expE) where only emissions were assimilated was not included here. But it was still
worth to simultaneously assimilate the chemical ICs and emission. For one thing, in
expE, the chemical concentrations can be updated by the WRF-Chem model
simulations with the assimilated emissions as the initial field in each DA cycle. That
means that the 50-member ensemble forecasts were performed twice and it was time
consuming. For another, better concentration analysis could be obtained in expJ due
to the simultaneous assimilation of ICs and emissions. While in expE, there may be
larger uncertainties for the updated chemical concentrations through WRF-Chem due
to the deficiency of chemistries and the uncertainties of the ICs. This will lead to
larger uncertainties for the emission inversion. Also the improvement of $PM_{2.5}$
forecasts will be limited due to the comparatively poor chemical ICs.

This study represents the first step in the simultaneous optimization of chemical

ICs and emissions and only surface $PM_{2.5}$ measurements were assimilated. In future
work, gas-phase observations of $SO_2$, $NO_2$ and CO will be used to further improve the
performance of this DA system.

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

Kahnert, M.: Variational data analysis of aerosol species in a regional CTM:
Background error covariance constraint and aerosol optical observation operators,
Tellus, Ser. B, 60, 753–770, doi:10.1111/j.1600-0889.2008.00377, 2008.
Kleist, D. T., Parrish, D. F., Derber, J. C., Treadon, R., Wu, W.-S., and Lord, S.:
Introduction of the GSI into the NCEP global data assimilation system, Weather
Forecast., 24, 1691–1705, 2009.
Huneeus, N., Chevallier, F., and Boucher, O.: Estimating aerosol emissions by
assimilating observed aerosol optical depth in a global aerosol model, Atmos.
Chem. Phys., 12, 4585-4606, doi:10.5194/acp-12-4585-2012, 2012.
Huneeus, N., Boucher, O., and Chevallier, F.: Atmospheric inversion of SO2 and
primary aerosol emissions for the year 2010, Atmos. Chem. Phys., 13,
6555-6573, doi:10.5194/acp-13-6555-2013, 2013.
Hunt, B., Kostelich, E., and Szunyogh, I.: Efficient data assimilation for
spatiotemporal chaos: a Local Ensemble Transfom Kalman Filter, Physica D,

230, 112–126, 2007.

Lee, E.-H., Ha, J.-C., Lee, S.-S., and Chun, Y.: PM10 data assimilation over South Korea to Asian dust forecasting model with the optimal interpolation method, Asia-Pacific J. Atmos. Sci., 49(1), 73–85, doi:10.1007/s13143-013-0009-y, 2013.

Li, Z., Zang, Z., Li, Q. B., Chao, Y., Chen, D., Ye, Z., Liu, Y., and Liou, K. N.: A three-dimensional variational data assimilation system for multiple aerosol species with WRF/Chem and an application to $PM_{2.5}$ prediction, Atmos. Chem. Phys., 13, 4265-4278, doi:10.5194/acp-13-4265-2013, 2013.

Li, J., Li, Y., Bo, Y., and Xie, S.: High-resolution historical emission inventories of crop residue burning in fields in China for the period 1990–2013, Atmos. Environ., 138, 152–161, 2016.

Liu, Z., Liu, Q., Lin, H. C., Schwartz, C. S., Lee, Y. H., and Wang, T.: Three-dimensional variational assimilation of MODIS aerosol optical depth: implementation and application to a dust storm over East Asia, J. Geophys. Res., 116, D23206, doi:10.1029/2011JD016159, 2011.

Liu, F., Zhang, Q., Tong, D., Zheng, B., Li, M., Huo, H., and He, K. B.: High-resolution inventory of technologies, activities, and emissions of coal-fired power plants in China from 1990 to 2010, Atmos. Chem. Phys., 15, 13299-13317, doi:10.5194/acp-15-13299-2015, 2015.

McLinden, C.A., Fioletov, V., Shephard, M.W., Krotkov, N., Li, C., Martin, R.V., Moran, M.D., and J. Joiner,: Space-based detection of missing sulfur dioxide sources of global air pollution, Nat. Geosci., 9, 496–500, doi:10.1038/ngeo2724, 2016.

Mijling, B. and van der A, R. J.: Using daily satellite observations to estimate emissions of short-lived air pollutants on a mesoscopic scale, J. Geophys. Res., 117, D17302, doi:10.1029/2012JD017817, 2012.

Miyazaki, K., Eskes, H. J., Sudo, K., Takigawa, M., van Weele, M., and Boersma, K. F.: Simultaneous assimilation of satellite $NO_2$, $O_3$, CO, and $HNO_3$ data for the analysis of tropospheric chemical composition and emissions, Atmos. Chem.

Phys., 12, 9545– 9579, doi:10.5194/acp-12-9545-2012, 2012.
Miyazaki, K., Eskes, H. J., Sudo, K., and Zhang, C.: Global lightning NOx production

estimated by an assimilation of multiple satellite data sets, Atmos. Chem. Phys.,

14, 3277–3305, doi:10.5194/acp-14-3277-2014, 2014.

Ott, E., Hunt, B. R., Szunyogh, I., Zimin, A. V., Kostelich, E. J., et al.: Exploiting

local low dimensionality of the atmospheric dynamics for efficient Kalman

filtering,        arXiv:physics/0203058,        24        pp.,        available        at:

http://arxiv.org/abs/physics/0203058v3/, 2002.

Ott, E., Hunt, B. R., Szunyogh, I., Zimin, A. V., Kostelich, E. J., et al.: A local

ensemble Kalman filter for atmospheric data assimilation, Tellus A, 56, 415–428,

2004.

Pagowski, M., Grell, G. A., McKeen, S. A., Peckham, S. E., and Devenyi, D.:

Three-dimensional variational data assimilation of ozone and fine particulate

matter observations: some results using the Weather Research and Forecasting –

Chemistry model and Grid-point Statistical Interpolation, Q. J. Roy. Meteor. Soc.,

136, 2013–2024, doi:10.1002/qj.700, 2010.

Pagowski, M., and Grell, G. A.: Experiments with the assimilation of fine aerosols

using an ensemble Kalman filter, J. Geophys. Res.-Atmos., 117, D21302,

doi:10.1029/2012jd018333, 2012.

Peng, Z., Zhang, M., Kou, X., Tian, X., and Ma, X.: A regional carbon data

assimilation system and its preliminary evaluation in East Asia, Atmos. Chem.

Phys., 15, 1087-1104, doi:10.5194/acp-15-1087-2015, 2015.

Pope, C. A.: Review: Epidemiological basis for particulate air pollution health

standards, Aerosol Sci. Tech., 32, 4–14, 2000.

Pope, C. A., Burnett, R. T., Thun, M. J., Calle, E. E., Krewski, D., Ito, K., and

Thurston, G. D.: Lung cancer, cardiopulmonary mortality, and long-term

exposure to fine particulate air pollution, J. Am. Med. Assoc., 287, 1132–1141,

2002.

Rubin, J. I., Reid, J. S., Hansen, J. A., Anderson, J. L., Collins, N., Hoar, T. J., Hogan,

988        T., Lynch, P., McLay, J., Reynolds, C. A., Sessions, W. R., Westphal, D. L., and

Zhang, J.: Development of the Ensemble Navy Aerosol Analysis Prediction
System (ENAAPS) and its application of the Data Assimilation Research
Testbed (DART) in support of aerosol forecasting, Atmos. Chem. Phys., 16,
3927-3951, doi:10.5194/acp-16-3927-2016, 2016.

Saide, P. E., Carmichael, G. R., Liu, Z., Schwartz, C. S., Lin, H. C., da Silva, A. M.,
and Hyer, E.: Aerosol optical depth assimilation for a size-resolved sectional
model: impacts of observationally constrained, multi-wavelength and fine mode
retrievals on regional scale analyses and forecasts, Atmos. Chem. Phys., 13,
10425-10444, doi:10.5194/acp-13-10425-2013, 2013.

Schwartz, C. S., Liu, Z., Lin, H. C., and McKeen, S. A.: Simultaneous
three-dimensional variational assimilation of surface fine particulate matter and
MODIS aerosol optical depth, J. Geophys. Res., 117, D13202,
doi:10.1029/2011JD017383, 2012.

Schwartz, C. S., Liu, Z., Lin, H.-C., and Cetola, J. D.: Assimilating aerosol
observations with a "hybrid" variational-ensemble data assimilation system, J.
Geophys. Res. Atmos., 119, 4043–4069, doi:10.1002/2013JD020937, 2014.

Sekiyama, T. T., Tanaka, T. Y., Shimizu, A., and Miyoshi, T.: Data assimilation of
CALIPSO aerosol observations, Atmos. Chem. Phys., 10, 39-49,
doi:10.5194/acp-10-39-2010, 2010.

Schutgens, N. A. J., Miyoshi, T., Takemura, T., and Nakajima, T.: Sensitivity tests for
an ensemble Kalman filter for aerosol assimilation, Atmos. Chem. Phys., 10,
6583-6600, doi:10.5194/acp-10-6583-2010, 2010.

Schutgens, N. A. J., Miyoshi, T., Takemura, T., and Nakajima, T.: Applying an
ensemble Kalman filter to the assimilation of AERONET observations in a
global aerosol transport model, Atmos. Chem. Phys., 10, 2561-2576,
doi:10.5194/acp-10-2561-2010, 2010.

Schutgens, N., Nakata, M., and Nakajima, T.: Estimating Aerosol Emissions by
Assimilating Remote Sensing Observations into a Global Transport Model,
Remote Sensing,4, 3528-3543, 2012.

Tang, X., Zhu, J., Wang, Z. F., and Gbaguidi, A.: Improvement of ozone forecast over

Beijing based on ensemble Kalman filter with simultaneous adjustment of initial conditions and emissions, Atmos. Chem. Phys., 11, 12901–12916, doi:10.5194/acp-11-12901-2011, 2011.

Tombette, M., Mallet, V., and Sportisse, B.: PM10 data assimilation over Europe with the optimal interpolation method, Atmos. Chem. Phys., 9, 57-70, doi:10.5194/acp-9-57-2009, 2009.

Torn, R. D., Hakim, G. J., and Snyder, C.: Boundary conditions for limited-area ensemble Kalman filters, Mon. Weather Rev., 134, 2490–2502, 2006.

van Loon, M., Builtjes, P. J. H., and Segers, A. J.: Data assimilation of ozone in the atmospheric transport chemistry model LOTOS, Environ. Model. Softw., 15, 603–609, 2000.

Wang, J., Xu, X., Henze, D. K., Zeng, J., Ji, Q., Tsay, S.-C., and Huang, J.: Top-down estimate of dust emissions through integration of MODIS and MISR aerosol retrievals with the GEOS-Chem adjoint model, Geophys. Res. Lett., 39, L08802, doi:10.1029/2012GL051136, 2012.

Wang, Y. X., Zhang, Q. Q., Jiang, J. K., Zhou, W., Wang, B. Y., He, K. B., Duan, F. K., Zhang, Q., Philip, S., and Xie, Y. Y.: Enhanced sulfate formation during China's severe winter haze episode in January 2013 missing from current models, J.Geophys.Res.-Atmos., 119, 10.1002/2013JD021426, 2014

Wang, X.Y., Liang,. X.Z., Jiang, W.M., Tao, Z.N., Wang, J.X.L., Liu, H.N., Han Z.W., Liu, S.Y., Zhang, Y.Y., Grell, G.A., Peckham, S.E.: WRF-Chem simulation of East Asian air quality: Sensitivity to temporal and vertical emissions distributions, Atmospheric Environment,44(2010) 660-669

Whitaker, J. S., and Hamill, T. M.: Ensemble data assimilation without perturbed observations, Mon. Weather Rev., 130, 1913–1924, 2002.

Woo, J.H., Baek, J.M., Kim, J.W., Carmichaael, G.R., Thongboonchoo, N., Kim, S.T., An, J.H.: Development of a Multi-Resolution Emission Inventory and Its Impact on Sulfur Distribution for Northeast Asia, Water, Air, and Soil Pollution 148: 259–278, 2003.

Weekly Crop Residue Burning Monitoring Report ,

http://hjj.mep.gov.cn/jgjs/201510/P020151012746205487305.pdf,     2015a     (in
Chinese).
Weekly       Crop       Residue       Burning       Monitoring       Report,
http://hjj.mep.gov.cn/jgjs/201510/P020151019568921489639.pdf,     2015b(in
Chinese).
Xia Y., Zhao, Y., Nielsen, C.P., Benefits of China's efforts in gaseous pollutant
control indicated by the bottom-up emissions and satellite observations
2000-2014, Atmospheric Environment, 136, 43-53, 2016
Yu, H., Dickinson, R. E., Chin, M., Kaufman, Y. J., Geogdzhayev, B., and
Mishchenko, M. I.: Annual cycle of global distributions of aerosol optical depth
from integration of MODIS retrievals and GOCART model simulations, J.
Geophys. Res., 108(D3), 4128, doi:10.1029/2002JD002717, 2003.
Yumimoto, K., Uno, I., Sugimoto, N., Shimizu, A., and Satake, S.: Adjoint inverse
modeling of dust emission and transport over East Asia, Geophys. Res. Lett., 34,
L00806, doi:10.029/2006GL028551, 2007.
Yumimoto, K., Uno, I., Sugimoto, N., Shimizu, A., Liu, Z., and Winker, D. M.:
Adjoint inversion modeling of Asian dust emission using lidar observations,
Atmos. Chem. Phys., 8, 2869-2884, doi:10.5194/acp-8-2869-2008, 2008.
Yumimoto, K., Nagao, T.M., Kikuchi, M., Sekiyama, T.T, Murakami, H.,Tanaka,
T.Y., Ogi, A., Irie, H., Khatri, P., Okumura, H., Arai, K., Morino, I., Uchino, O.,
Maki, T.: Aerosol data assimilation using data from Himawari-8, a
next-generation geostationary meteorological satellite, Geophys. Res. Lett., 43,
5886–5894, 2016.
Yin, X.M., Dai, T., Xin, J.Y., Gong, D.Y., Yang, J., Teruyuki, N., Shi, G.Y.:
Estimation of aerosol properties over the Chinese desert region with MODIS
AOD assimilation in a global model, Adv. Clim. Change Res., 7, 90–98, 2016.
Zhang, J., Reid, J. S., Westphal, D., Baker, N., and Hyer, E.: A System for
Operational Aerosol Optical Depth Data Assimilation over Global Oceans, J.
Geophys. Res., 113, D10208, doi:10.1029/2007JD009065, 2008.
Zhang, Q., Streets, D. G., Carmichael, G. R., He, K. B., Huo, H., Kannari, A.,
Klimont, Z., Park, I. S., Reddy, S., Fu, J. S., Chen, D., Duan, L., Lei, Y., Wang,
L. T., and Yao, Z. L.: Asian emissions in 2006 for the NASA INTEX-B mission,
Atmos. Chem. Phys., 9, 5131-5153, doi:10.5194/acp-9-5131-2009, 2009.
Zhang, L., Liu, L. C., Zhao, Y. H., Gong, S. L., Zhang, X. Y., Henze, D. K., Capps, S.
L., Fu, T. M., Zhang, Q., and  Wang, Y. X.: Source attribution of particulate
matter pollution over North China   with the   adjoint method, Environ.Res.Lett.,
10, Artn 08401110.1088/1748-9326/10/8/084011, 2015.
Zheng, B., Zhang, Q., Zhang, Y., He, K. B., Wang, K., Zheng, G.
J., Duan, F. K., Ma, Y. L., and Kimoto, T.: Heterogeneous
chemistry: a mechanism missing in current models to explain
secondary inorganic aerosol  formation during the January 2013 haze
episode in North China,     Atmos.Chem.Phys.,    15,    2031-2049,
10.5194/acp-15-2031-2015, 2015.



Figure 9. Time series of the hourly PM2.5 obtained from observations (circle),
analysis (blue line), control run (black line) and hourly output of 48-h forecast in three
megacities: (a) Beijing; (c) Shanghai; and (e) Guangzhou in expJ and (b) Beijing; (d)
Shanghai; and (f) Guangzhou in expC. See text in section 5.4.

Figure 10. Bias of surface PM2.5 as a function of forecast range calculated against
independent observations over the three sub-regions: (a) Beijing–Tianjin–Hebei
region; (c) Yangtze River delta; (e) Pearl River delta and RMSE over (b) Beijing–
Tianjin–Hebei region; (d) Yangtze River delta; (f) Pearl River delta; (g) Normalized
RMSE (assimilation divided by control) for expJ and (h) (g) Normalized RMSE for
expC.




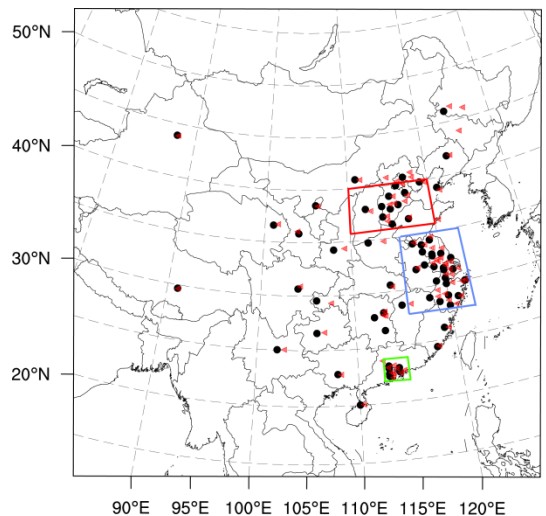


Figure 1. Locations of 77 PM$_{2.5}$ assimilation observation stations (black dot) and the
77 independent observation stations (red triangle) in the model domain. The three
colored boxes mark sub-regions with relatively dense coverage for the Beijing–
Tianjin–Hebei region (JJJ, 12 assimilation stations and 12 independent stations, red
box), the Yangtze River delta (YRD, 24 assimilation stations and 24 independent
stations, blue box) and the Pearl River delta (PRD, 9 stations and 9 independent
stations, green box).


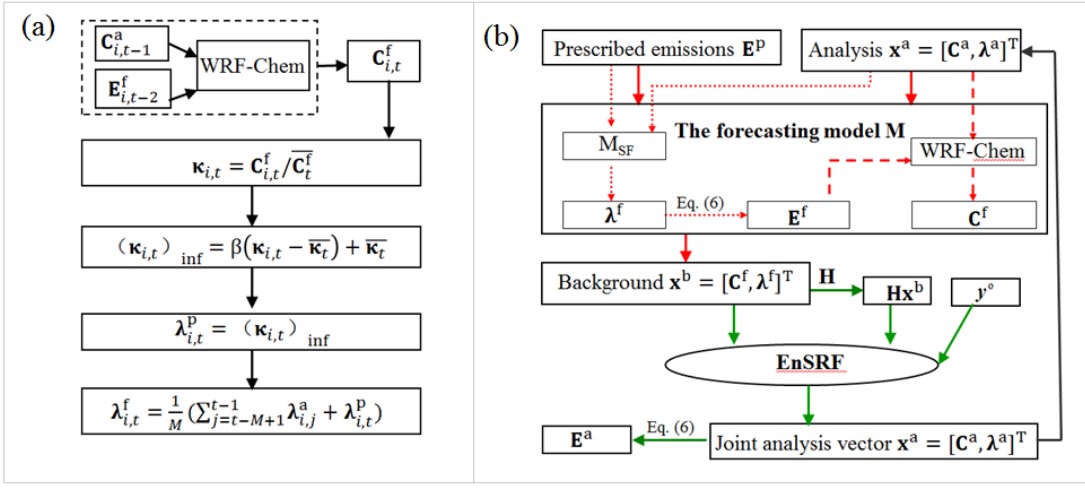


Figure 2. (a) Framework of $\mathbf{M}_{SF}$ and (b) flow chart of the data assimilation system
that simultaneously optimizes the chemical initial conditions and emissions.


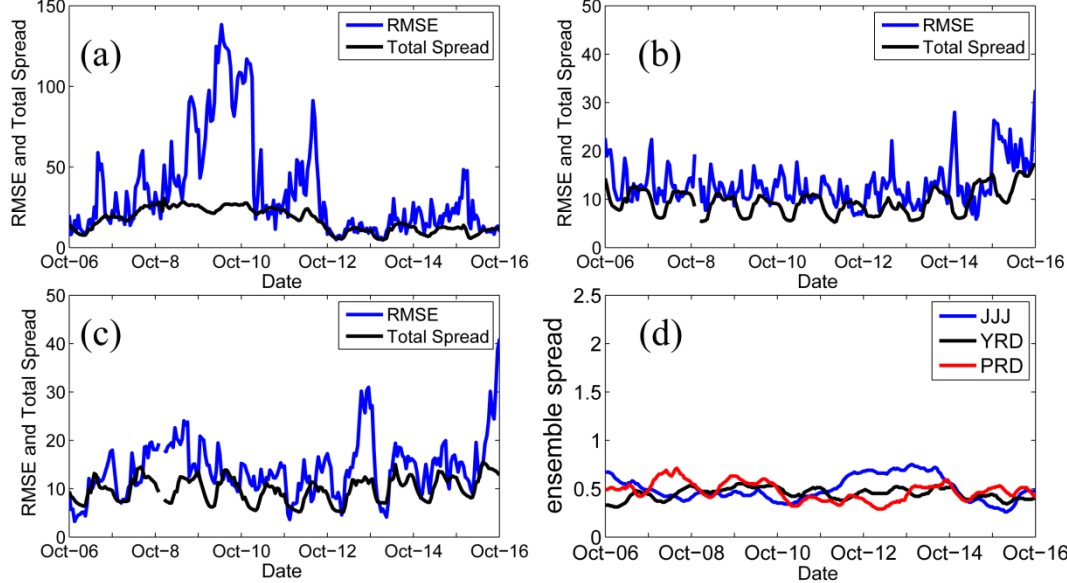


Figure 3. Time series of prior ensemble mean RMSE and total spread for $PM_{2.5}$
concentrations aggregated over all observations over the three sub-regions: (a)
Beijing–Tianjin–Hebei region; (b) Yangtze River delta; (c) Pearl River delta; and (d)
time series of the area mean ensemble spread for $\lambda_{PM2.5}$ over the three sub-regions.

Table 1. Comparison of the surface PM$_{2.5}$ mass concentrations from the control and
assimilation experiments to observations over all analysis times from 6 to 16 October

2014.

| Region | Experiment | Mean observed value | Mean simulated value | BIAS | RMSE | CORR |
|---|---|---|---|---|---|---|
| Beijing– | Control | | 98.3 | −18.0 | 81.6 | 0.790 |
| Tianjin– | expJ | 116.3 | 106.0 | −10.3 | 66.9 | 0.827 |
| Hebei | expC | | 104.1 | -12.2 | 64.0 | 0.845 |
| Yangtze | Control | | 64.4 | 15.9 | 30.6 | 0.593 |
| River | expJ | 48.5 | 46.9 | -1.6 | 15.3 | 0.846 |
| delta | expC | | 46.1 | -2.4 | 17.3 | 0.803 |
| Pearl | Control | | 82.4 | 20.6 | 31.8 | 0.624 |
| River | expJ | 61.8 | 66.5 | 4.7 | 16.1 | 0.800 |
| delta | expC | | 64.1 | -2.3 | 15.6 | 0.797 |



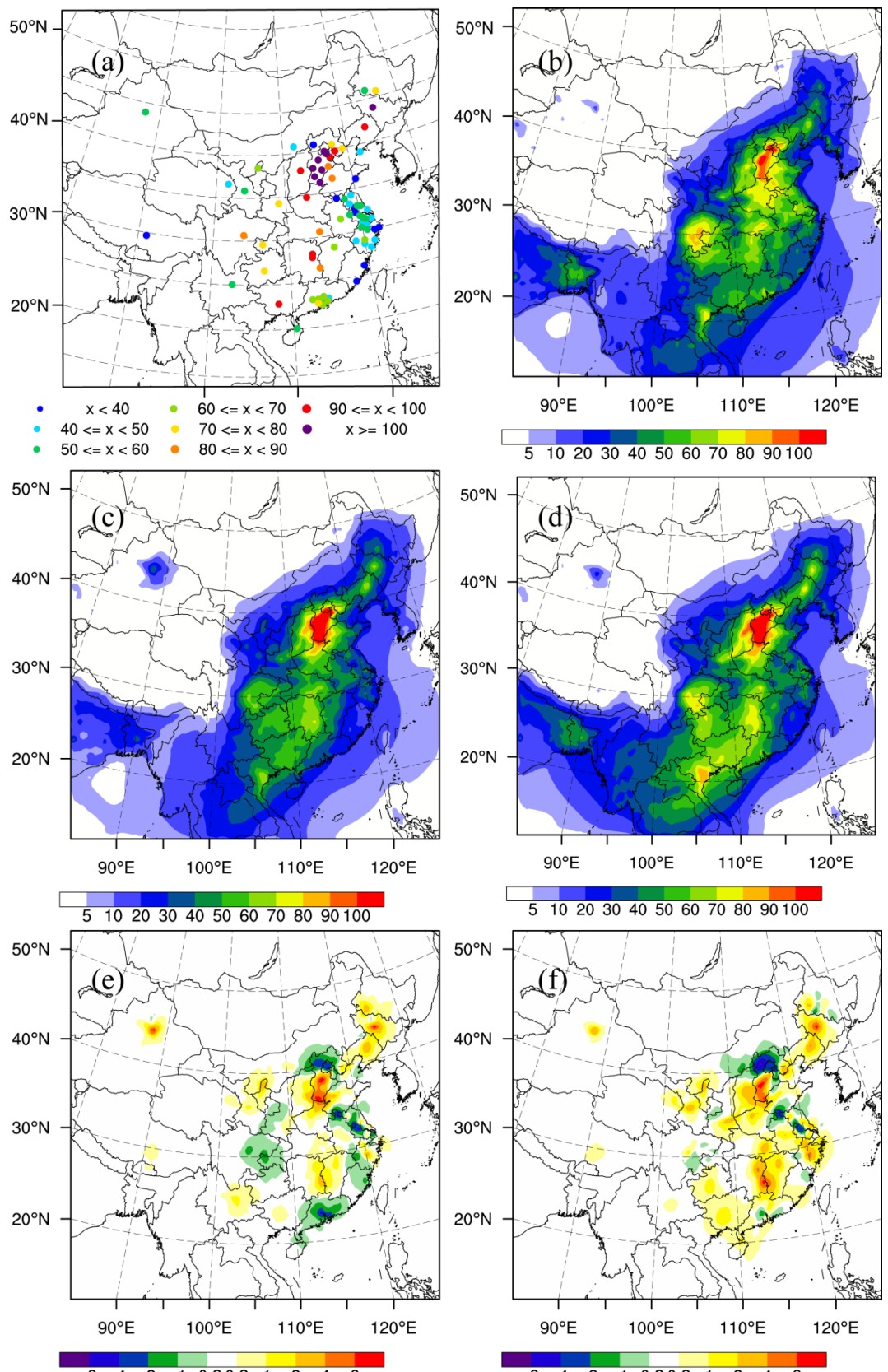

Figure 4. Spatial distribution of the PM$_{2.5}$ mass ($\mu$g m$^{-3}$) of the (a) observations; (b) simulation of the control run; (c) analysis of expJ; (d) analysis of expC; (e) increments of expJ; (f) increments of expC; at the lowest model level averaged over all hours

 from 6 to 16 October 2014.

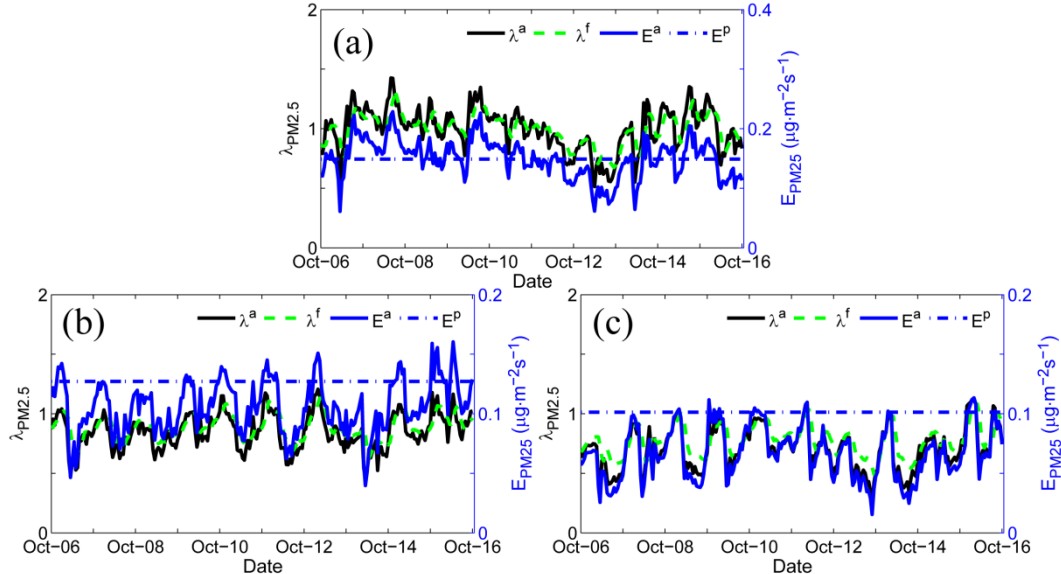

Figure 5. Hourly area-averaged time series of emission scaling factors (black) extracted from the ensemble mean of the analyzed $\boldsymbol{\lambda}_{\text{PM2.5}}^{\text{a}}$ and the corresponding analyzed unspeciated primary PM$_{2.5}$ emissions $\mathbf{E}_{\text{PM2.5}}^{\text{a}}$ (blue) over the three sub-regions: (a) Beijing–Tianjin–Hebei region; (b) Yangtze River delta; and (c) Pearl River delta.

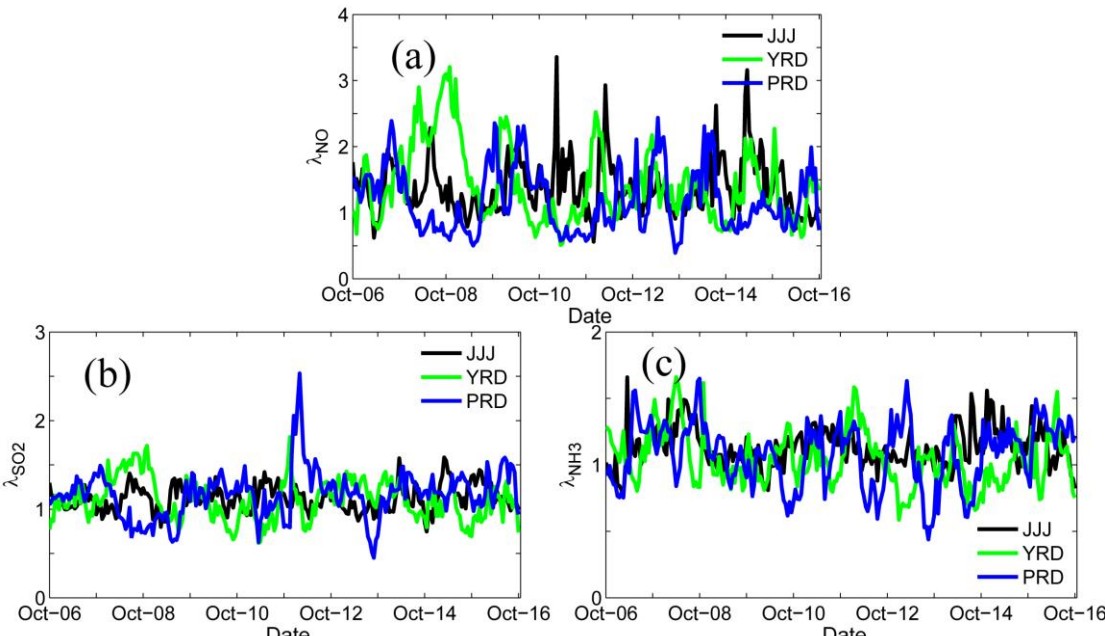


Figure 6. Hourly area-averaged time series of emission scaling factors extracted from
the ensemble mean of the analyzed (a) $\lambda_{NO}^{a}$; (a) $\lambda_{SO2}^{a}$; (a) $\lambda_{NH3}^{a}$ over the three
sub-regions: Beijing–Tianjin–Hebei region (JJJ, black), Yangtze River delta (YRD,
green), and Pearl River delta (PRD, blue).


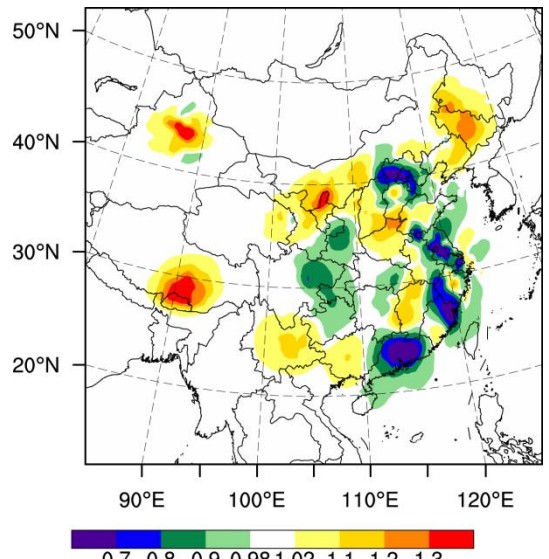

Figure 7. Spatial distribution of $\lambda_{PM2.5}$ at the lowest model level averaged over all hours from 6 to 16 October 2014.


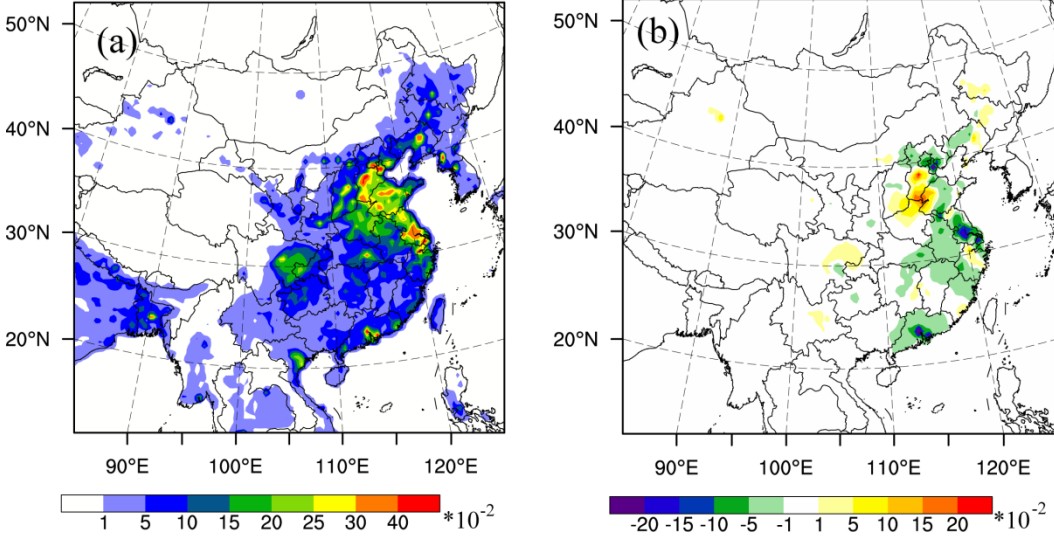


Figure 8. Spatial distribution of (a) the prior unspeciated primary sources of PM$_{2.5}$
(μg m$^{-2}$ s$^{-1}$) and (b) the time-averaged differences between the ensemble mean
analysis and the prior values (μg ·m$^{-2}$ s$^{-1}$) at the lowest model level averaged over all
hours from 6 to 16 October 2014.


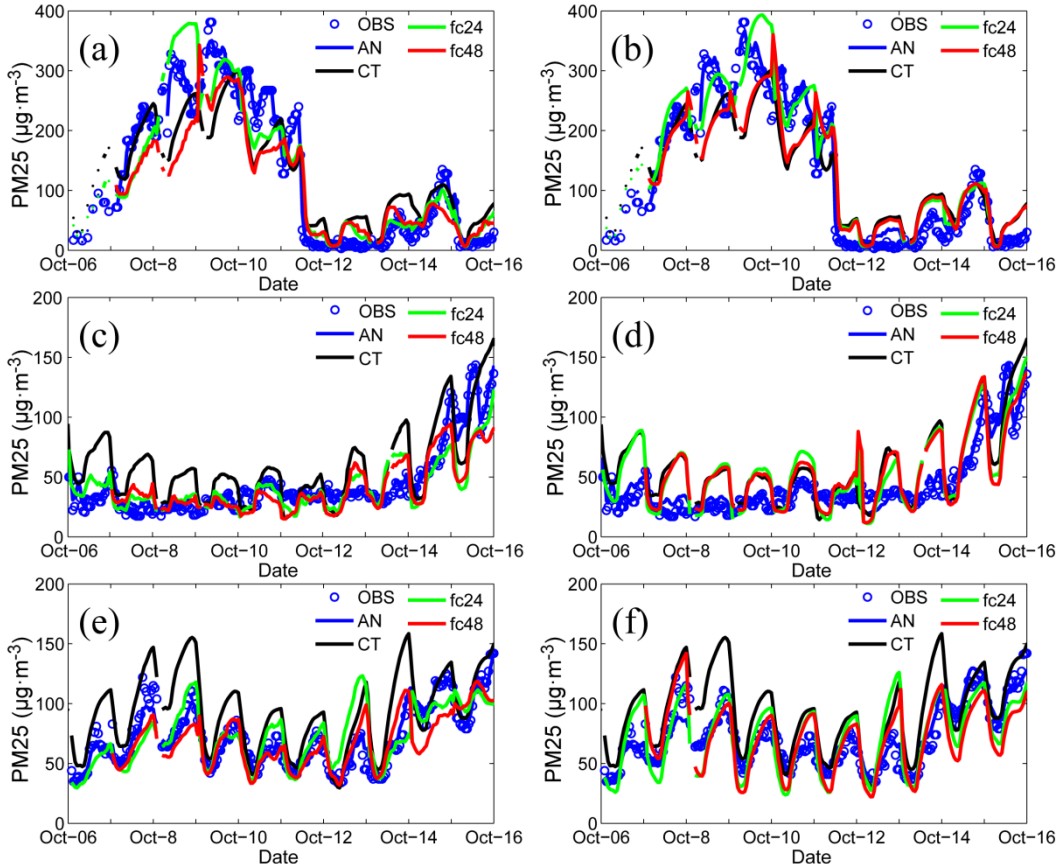


Figure 9. Time series of the hourly PM$_{2.5}$ obtained from observations (circle), analysis

(blue line), control run (black line) and hourly output of 48-h forecast in three

megacities: (a) Beijing; (c) Shanghai; and (e) Guangzhou in expJ and (b) Beijing; (d)

Shanghai; and (f) Guangzhou in expC. See text in section 5.4.




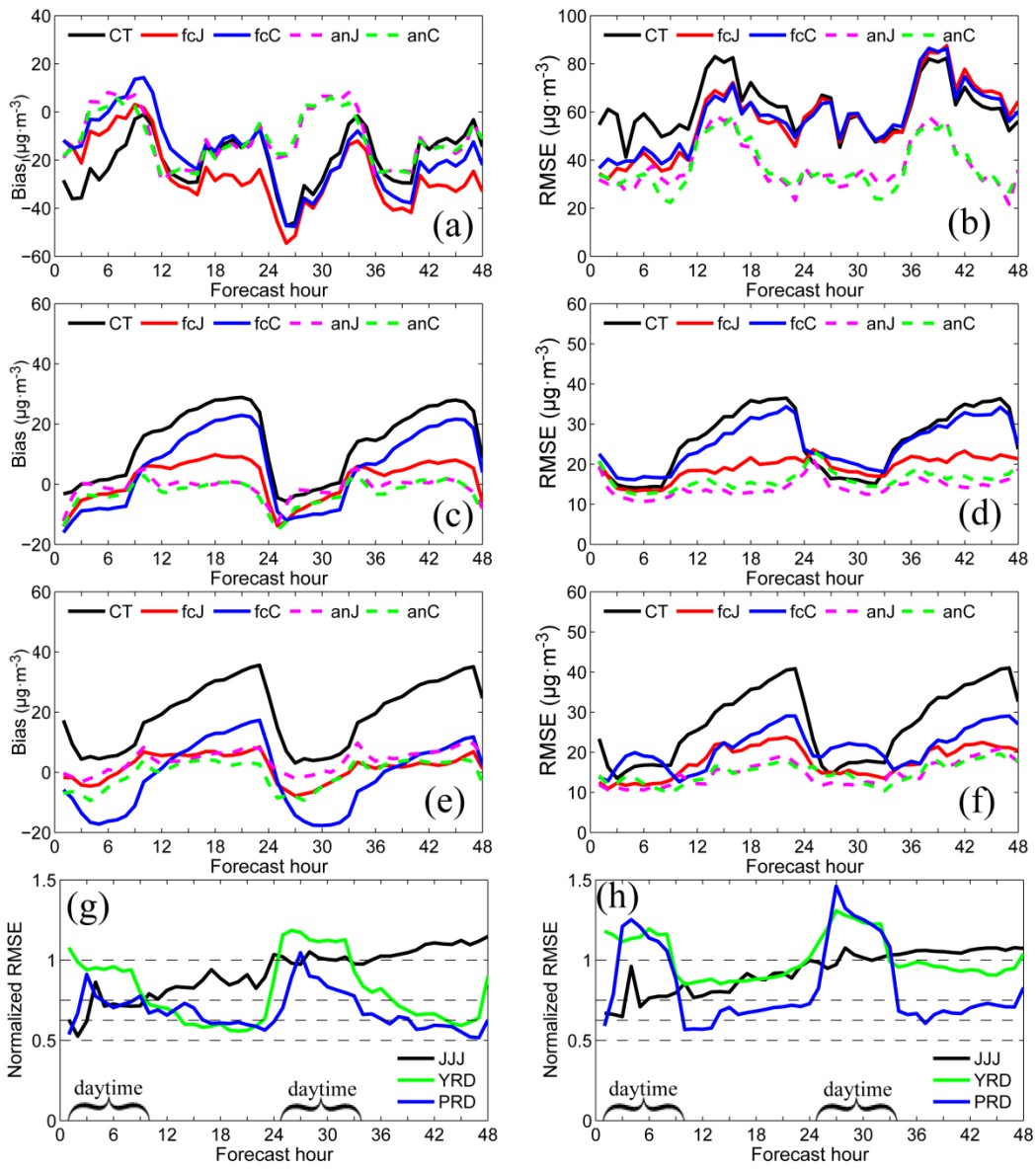


Figure 10. Bias of surface PM$_{2.5}$ as a function of forecast range calculated against all
the independent observations over the three sub-regions shown in figure 1: (a)
Beijing–Tianjin–Hebei region; (c) Yangtze River delta; (e) Pearl River delta and
RMSE over (b) Beijing–Tianjin–Hebei region; (d) Yangtze River delta; (f) Pearl
River delta; (g) Normalized RMSE (assimilation divided by control) for expJ and (h)
(g) Normalized RMSE for expC. The 48-h forecasts were performed at each 0000
UTC from 6 to 16 October 2014 and the statistics were computed from 6 to 16
October.
