# Peer review of "Improving PM2.5 forecast over China by the joint adjustment of initial conditions"

_Atmospheric Chemistry and Physics, 2016_

## Referee Comment (RC1) · Anonymous Referee #1 · 26 Sep 2016

The author extends the EnSRF algorithm to adjust the chemical initial conditions (ICs) and the pollutant emissions to obtain better forecasting of surface PM2.5 over China. They compare analyzed and forecasted PM2.5 concentration fields with the control simulation that has no data assimilation, and find that optimized initial condition and emissions achieves much better forecasting for the 48-h forecasts. Their results suggest that the joint adjustment (not only ICs but also emissions) contributes the substantial improvement in the longer (34–48 hours) forecasts. Their investigation is interesting and valuable. The manuscript is well written and structured. I recommend publication after addressing the following concerns.

General comments:

[Figure]

1:The authors suggest that the joint adjustment (initial conditions and emissions) provides substantial improvements in from 34- to 48-h forecasts. Do you perform an assimilation and forecasting experiment in which only ICs are adjusted. Comparing between results from the joint adjustment and the IC only adjustment will reinforce your suggestion.

2:Both analyzed and forecasting results are validated by only observations that used in the assimilation. You should include the independent data, which is not used in the observational constraint, in the validation.

Specific comments:

3: Line 40, There are more recent research papers of ensemble-based assimilations with observations derived from in-situ measurements and geostationary satellite.

Dai, T., et al. (2014) Improvement of aerosol optical properties modeling over Eastern Asia with MODIS AOD assim- ilation in a global non-hydrostatic icosahedral aerosol transport model, Environ. Pollut., 195, 319–329.

Ying, X.M., et al. (2016) Estimation of aerosol properties over the Chinese desert region with MODIS AOD assimilation in a global model, Adv. Clim. Change Res., 7, 90–98.

Yumimoto, K., et al. (2016), Aerosol data assimilation using data from Himawari-8, a next-generation geostationary meteorological satellite, Geophys. Res. Lett., 43, 5886–5894.

4: Line 90, Does the observation operator (H) include function (conversion) for the emission scaling factor (lambda) or, in other words, does the lambda directly affect the model results in the observation state (Hx) through the observation operator? If no, how does the observations adjust the emission scaling factors in the assimilation process?

5: Line 139, The ensemble concentration ratio (Kappa) is defined by concentrations

of the ensemble forecasting. Can you confirm that the ensemble concentration ratio is random and the ensemble mean of Kappa becomes 1?

6: Line 152 or Equation (10), The denominator in the right hand should be 1/M+1?

7: Line 183, As shown in Equation (12), dust and sea salt aerosols can contribute PM2.5 concentrations. Do you include emissions of dust and sea salt in the assimilation process?

8: Line 190, A period may drop in the end of state.

9: Figure 1, Could you check figure 1 again? Some characters and numbers of equation are different from those in the manuscript.

10: Line 202, Does this means that you need to perform the 50-member ensemble forecast twice in your assimilation system?

11: Line 254, How often did this exclusion occur? Figure 3a and 8a imply that quite a few large departures occurs in the JJJ region during 9–10 October.

12: Line 281, How do you decide the ensemble member of 50?

13: Line 349, Could you add mean distribution of PM2.5 concentration from the control and assimilation simulations in Figure 4? These will make the reader to understand a priori distribution and the adjustment of PM2.5 concentrations easily. Plotting mean observed PM2.5 concentrations on these map will be even better.

14: Line 349, We can find adjustments over the SE Asia and India where you have no PM2.5 observation.

15: Figure 5, Overlaying of a priori emissions (it will be flat lines) in Figure 5 may emphasize that the assimilation can generate the temporal variations in the emissions.

16: Line 375, Is the burning of crop residues limited in the JJJ region? Li et al. shows that the northern part of YRD also has large emissions from the burning.

17: Line 382, Do you confirm that temporal and horizontal distributions of a posteriori gaseous emissions are reasonable? Could you show temporal distribution of temporal variation of gaseous emissions (like Figure 5 and Figure 6)?

18: Figure 6, We can find there are adjustments of lambda over the ocean where we have no PM2.5 emission (Figure 7a). Do you define the lambda over the ocean?

19: Line 395, "less constraint on the sources of the secondary aerosol precursors" means that the adjustments of emissions of the secondary aerosol precursors have little effect on PM2.5 forecasting?

20: Line 467, The exclusion due to large discrepancy between first guess and observed concentrations may be partly responsible for the sparse observational constraint in the JJJ region during the heavy PM2.5 event.

---

## Referee Comment (RC2) · Anonymous Referee #2 · 19 Oct 2016

**Improving PM2.5 forecast over China by the joint adjustment of initial conditions and source emissions with an ensemble Kalman filter**

Zhen Peng, Zhiquan Liu, Dan Chen, Junmei Ban

The authors introduce a DA system based on an ensemble square root filter combined with WRF-Chem that assimilates surface observations of PM2.5 across China. The novelty is that they use both aerosol concentrations and emissions in their DA state vector (although it should be noted they did something very similar for CO2 in Peng et al ACP 2015).

While the main idea is interesting and the topic is certainly relevant to ACP, I recommend against publication for the following reasons: 1) no independent observations are used to evaluate results. While this is ok for the evaluation of forecasts, this is not good practice for the evaluation of analyses; 2) no proof is offered for the central contention that analyzing emissions *together with* concentrations improves results; 3) no proof is offered for the second central contention that this system improves emissions; 4) many assumptions are merely stated without due reference, deliberation or any kind of sensitivity study; 5) several conclusions are drawn based on irrelevant data (see my comments).

It should be noted that reviewer 1 mentions the first two points as well but is apparently more lenient.

Point 3 I find particularly important as this is a contention made by other authors as well (Tang et al, Miyazaki et al) with little in the form of proof. Models have errors, and analysing emissions may simply balance out some of these errors without improving the emissions. Note that we do no have observations to evaluate those emissions but this can not be used as an argument to forego proper scientific reasoning.

In addition I find the structure of the paper illogical, and missed important information on details of their DA system and several references to previous attempts at emission estimation.

I hope the authors will continue this work but put more effort in stating their case convincingly, for this research topic is certainly worthwhile. Maybe my comments can be of some help towards improving this manuscript.

**Abstract**

P 1, L 13: "The forecast model of emission scaling factors was developed by associating the time smoothing operator with WRF-Chem forecast chemical concentrations". Please rephrase, this sentence is hard to understand without reading the paper first.

**Introduction**

P 2, L 40: The authors seem unaware of a lot of previous work on ensemble-based DA: Sekiyama et al ACP 2010, Schutgens et al. ACP 2010a, Schutgens et al ACP 2010b. , Dai et al, *Env. Pol.* 2014,  Rubin et al. ACP 2016, , Yumimoto et al GRL 2016. Please include those references

P 2,  L 50: Again, several references seem to be missing i.c. emission estimation. For aerosol: Zhang et al JGR 2005, Sekiyama et al. ACP 2010, Huneeus et al ACP 2012, Schutgens et al. Rem Sens 2012, Huneeus et al ACP 2013

**Methodology**

P 3, L 78: Please introduce the ENSRF in context of some other EnKF (EAKF, LEKF, LETKF). What is the reason for this choice of EnKF, what is it main strength/weakness?

P  54 L 94: Change "can be approximated" to "will be approximated". It is by no means certain that this is a good approximation. Part of the evaluation & tuning of an EnKF involves exactly the sampling errors introduced by Eq 5 & 6

P 3: Since the DA depends on the forecast model's details, I suggest to first discuss the forecast model (and introduce **C** and  **λ,** and only then the ENSRF)

P 4, L 105: Please provide a bit more information on the base setup of the model: domain size, grid resolution, major aerosol species

P 4, L 106: "to forecast the emission scaling factors and the aerosol control variables". What are the control variables? I guess the authors mean aerosol concentrations, please change this. Note that both **C** and **λ** form the state vector.

P 5, L 123: "for the lowest eight vertical levels": so the emission inventory included heights at which the emissions were injected? These heights are all within the boundary layer? Why are only the lowest 8 layers considered?

P 6, L 139: "$\kappa_{i,t}$ are random". I wouldn't call them random. I realize they are distributed around the mean $\bar{\bar{\kappa}}_t$, , but they were calculated through a short-term forecast of WRF-Chem.

P 6, L 144: "$\beta$ = 1.5 was chosen in this study": This sounds like an arbitrary choice? Normally $\beta$ results from tuning a DA but no such exercise was done?

P 6, L 145: "As the concentrations were closely related to the emissions": if I assume this refers to emissions and concentrations in the same grid-box (given the mathematics of their DA system), this is a bold statement and needs some strong arguments. I can see that during the dust season, Beijing area will be heavily impacted by dust from Eastern China, invalidating your assumptions. Even for pollution emissions, transport may actually be very important.

P 6, L 147: "concentration ratios ($\kappa_{i,t}$)inf served as the prior emission scaling factors $\lambda_{i,t}$" So the concentrations themselves were not inflated, as is usually done in EnKF? What is the justification for this? Shouldn't the scaling factors be perturbed according to the uncertainty in emission inventories and parametrisations?

P 6, L 152: I suspect that Eq 10 is missing a factor 0.5. The prior and analysis scale factors are previous times are averaged.

P 6, L 153: Again, a rather arbitrary choice (M=4)? How does this relate to the DA cycle?

P 6, L 159: "emission inventories". Except in the case of dust, sea-salt etc. Or are these not perturbed? If not, why are they not perturbed (surely they are uncertain as well)? Actually, the authors are rather sparse in their information. Is each species perturbed independently from the others? What is the level of perturbation? Are neighbouring grid-points perturbed independently or do you assume correlations?

P 7, L 175: "the state variables of the analysis of the ICs were the 15 WRF-Chem/GOCART aerosol variables." This should have been mentioned earlier, maybe line 101.

P 7, L 184: "($\lambda$PM2.5, $\lambda$SO2, $\lambda$NO and $\lambda$NH3)" This line and the following paragraph suggest that the authors keep the $E_{EC}$ and $E_{ORG}$ constant? They do not matter? I rather think they do. By the way, this paragraph might be rewritten to improve readability.

P 8, L 208: The authors never explain how the system is started up. Some initial perturbation in concentrations and/or emissions must be assumed.

P 9, L 247: "$\varepsilon_r = r\varepsilon_0\sqrt{\Delta x}/L$," Can the authors provide a reference for this form of the representation error? Why do they choose L=3 km? How can it be that the representation error is a function of the measurement error? These are two independent error sources.

P 9, L 252-255: Some statistics on how often this happened would be appreciated.

P 10, L 261: "The horizontal grid spacing was 40.5 km and there were
262 57 vertical levels with the model top at 10 hPa." This sort of information should be in Sect 2.2.1

P 1, L 265: "initialization and spin-up procedures" Please briefly state the spin-up procedure. For how long was the ensemble run before the first DA happened?

P11, L 279: "clean oceanic conditions." Does this mean that over land you assumed seasalt aerosol as LBC?

P 11, L 280: "standard Gaussian random noise". Please briefly state what standard deviations you assumed, and how you dealt with negative emissions.

P 13, L 336: "These statistics were calculated against observations over all the analyses" If I understand the authors, the same observations that were assimilated are here used to evaluate the results. This likely explains the high correlations. The authors should make it clear this is not an independent evaluation but merely a sanity check.

P 13, L 356: "These results indicate that DA greatly improved the ICs." This is rather bold as you have not used independent observations to evaluate the ICs. Obviously, if you nudge the model towards observations, the model will do better. Please remove this sentence.

P 13, L 363: "the optimized PM2.5 scaling factor, $\lambda$PM2.5a, showed an obvious variation with time, as did the optimized unspeciated primary sources of PM2.5, **E**PM2.5a" From the authors explanation of how their system works, I do not understand why $\lambda$PM2.5 and **E**PM2.5 would have a different (if only slightly) time evolution. Is this because they are regional averages?

P 13, L 379: "as the system is optimized based on ambient concentrations in which the transport and transformation processes are not directly taken into account" But surely transport is important? Maybe a Kalman smoother would have been a better system to solve this problem.

P 14, L 388: "at the lowest model level" Why do you only discuss emissions at lowest level? Are they much larger than those at higher levels? Surely it is the vertically integrated emissions that is important for the amount of particulate matter entering the atmosphere?

P 15, L 406: "Our assimilated PM2.5 and NOx emissions were in good agreement with this trend". The DA experiments reported here cover a period of a few weeks, so how can you compare that to a trend over 15 years?

P 17, L 470: "However, these results are still better than those obtained with the pure adjustment of ICs that lead to improvements in the first 12-h forecasts (Jiang et al., 2013; Schwartz et al., 2014)." This conclusion is baseless as Jiang et al use a different DA system (3D-VAR) with different observations (PM10) and Schwartz et al use a different domain (USA).

Figure 1: What is **F**? How is it related to Eq 1?

---

## Author Comment (AC1) · 2 Dec 2016

**Response to Reviewer #1's comments:**

We thank Referee # 1 for their thoughtful comments and suggestions that have helped to improve this manuscript. Our responses to comments (in bold style) and the corresponding changes to the manuscript are detailed below.

**General Comments:**

**1: The authors suggest that the joint adjustment (initial conditions and emissions) provides substantial improvements in from 34- to 48-h forecasts. Do you perform an assimilation and forecasting experiment in which only ICs are adjusted. Comparing between results from the joint adjustment and the IC only adjustment will reinforce your suggestion.**

We have performed other two experiments, the assimilation of pure chemical ICs and the corresponding 48-h forecasts experiment. The details are in the revised manuscript (Lines 432 to 434, Page 16; Lines 448 to 452, Page 17; Lines 513 to 533, Page 19; Lines 620 to 622, Page 23; Lines 665 to 704, Page 25).

**2: Both analyzed and forecasting results are validated by only observations that used in the assimilation. You should include the independent data, which is not used in the observational constraint, in the validation.**

We have used the independent observations to evaluate both the analyses and the forecasts. Please see the details in the revised manuscript (Lines 354 to 355, Page 13; Lines 502 to 515, Page 19; Lines 632, Page 23 to Lines 691, Page25).

**Specific comments:**

**3: Line 40, There are more recent research papers of ensemble-based assimilations with observations derived from in-situ measurements and geostationary satellite.**

**Dai, T., et al. (2014) Improvement of aerosol optical properties modeling over Eastern Asia with MODIS AOD assim- ilation in a global non-hydrostatic icosahedral aerosol transport model, Environ. Pollut., 195, 319–329.**

**Ying, X.M., et al. (2016) Estimation of aerosol properties over the Chinese desert**

**region with MODIS AOD assimilation in a global model, Adv. Clim. Change Res., 7, 90–98.**

**Yumimoto, K., et al. (2016), Aerosol data assimilation using data from Himawari-8, a next-generation geostationary meteorological satellite, Geophys. Res. Lett., 43, 5886–5894.**

We have added those references in Lines 47 to 48, Page 2.

**4: Line 90, Does the observation operator (H) include function (conversion) for the emission scaling factor (lambda) or, in other words, does the lambda directly affect the model results in the observation state (Hx) through the observation operator? If no, how does the observations adjust the emission scaling factors in the assimilation process?**

In this manuscript, the emission scaling factor $\boldsymbol{\lambda}^{\mathrm{f}}$ is calculated by the persistence forecasting operator $\mathbf{M}_{\mathrm{SF}}$. Then, the emissions are calculated using equation (6) (original Eq. 11). After that, the chemical fields $\mathbf{C}^{\mathrm{f}}$ are forecasted though WRF-Chem. Finally, the model-simulated $PM_{2.5}$ concentration at the observation space is calculated via equation (13) (original Eq. 12) (See details in Section 2.3.1). Therefore, $\boldsymbol{\lambda}^{\mathrm{f}}$ directly affect the model results.

In fact, for the adjustment of the emission scaling factors, $\mathbf{M}_{\mathrm{SF}}$ serves as the forecast model and the observation operator reflects the combined information of emissions (in the format of $\boldsymbol{\lambda}$ in equation (6)), the physics and chemistry processes in WRF-Chem simulations and the transformation of $PM_{2.5}$ from model space to observation space (equation (13)). We have addressed these in Lines 275 to 279, Page 11.

**5: Line 139, The ensemble concentration ratio (Kappa) is defined by concentrations of the ensemble forecasting. Can you confirm that the ensemble concentration ratio is random and the ensemble mean of Kappa becomes 1?**

The ensemble mean of the concentration ratio is $\overline{\kappa_{i,t}} = \overline{\mathbf{C}_{i,t}^{\mathrm{f}}/\overline{\mathbf{C}_t^{\mathrm{f}}}} = \overline{\mathbf{C}_{i,t}^{\mathrm{f}}}/\overline{\mathbf{C}_t^{\mathrm{f}}} =$

$\overline{C_t^f}/\overline{C_t^f}$=1. We have moved away random variables and revised this sentence in Line 142, Page 5.

**6: Line 152 or Equation (10), The denominator in the right hand should be 1/M+1?**

No. In Equation (5) (original Eq. 10), j starts from t-M+1. Thus, M scale factors (the prior and M-1 analysis scale factors) are used to calculate $\lambda_{i,t}^f$. For example, in our manuscript, M = 4. Thus, $\lambda_{i,t}^p$, $\lambda_{i,t-1}^a$, $\lambda_{i,t-2}^a$, and $\lambda_{i,t-3}^a$ are used. Therefore, the denominator in the right hand of Equation (5) is 1/M.

**7: Line 183, As shown in Equation (12), dust and sea salt aerosols can contribute PM2.5 concentrations. Do you include emissions of dust and sea salt in the assimilation process?**

We did not include emissions of dust and sea salt in the assimilation process as our focus is on the major anthropogenic emissions in mega-cities in China.

Emissions of dust and sea salt were parameterized within the GOCART model (Chin et al., 2002). Unlike the approach for anthropogenic emissions, the approach would be different to assimilate dust and sea salt. In addition, only the $PM_{2.5}$ measurements were used in this DA experiment, with such limited observations adding more control variables would cause much more uncertainties in the system which might lead to unreasonable analysis. This is our first attempt to improve $PM_{2.5}$ forecast by the joint adjustment of ICs and source emissions, so we primarily focus on the major anthropogenic sources in heavy polluted regions $(\mathbf{E}_{PM2.5i}, \mathbf{E}_{PM2.5j}, \mathbf{E}_{SO4i}, \mathbf{E}_{SO4j}, \mathbf{E}_{NO3i}, \mathbf{E}_{NO3j})$. Those emissions have large impacts on the distribution of $PM_{2.5}$, thus are updated in our analysis. In future work, more species of emissions might be included.

We have added some explanations in Lines 300 to 308, Page 12.

**8: Line 190, A period may drop in the end of state.**

I have revised the text Line 283, Page 11.

**9: Figure 1, Could you check figure 1 again? Some characters and numbers of equation are different from those in the manuscript.**

I have revised the figure.

**10: Line 202, Does this means that you need to perform the 50-member ensemble forecast twice in your assimilation system?**

No, we perform the forecast only once. The steps in this workflow are: (1) $\lambda_{PM2.5}^{f}$, $\lambda_{SO2}^{f}$, $\lambda_{NO}^{f}$ and $\lambda_{NH3}^{f}$ are calculated using the forecast chemical concentration fields of the previous assimilation cycle; (2) The ensemble members of the emissions are generated;(3) WRF-Chem forecasts the chemical fields; (4) EnSRF assimilates, at this step, the scaling factors and the chemical fields are assimilated; (6) the emissions are updated. So, WRF-Chem runs to forecast only once during a DA cycle.

I have mentioned this in Line 200, Page 8.

**11: Line 254, How often did this exclusion occur? Figure 3a and 8a imply that quite a few large departures occurs in the JJJ region during 9–10 October.**

The numbers of the observations were about 17700. Among them 8 observations were discarded because they were larger than 800 μg m$^{-3}$ and 243 (around 1.5%) were discarded due to the ensemble mean of the first guess departure exceeding 100 μg m$^{-3}$. In those 243 discarded observations, only 93 were in JJJ.

Figure 3a implied that some ensembles of the PM$_{2.5}$ background may deviate much from the observations during 9–10 October. However, the ensemble mean of the background PM$_{2.5}$ and the ensemble mean of the analysis PM$_{2.5}$ in the assimilation experiments were comparatively near to the observations (see ReFig1.), though the forecast of the PM$_{2.5}$ deviated much from the observations in the CT run and the forecast run. So only a few data were discarded due to the first guess departure

exceeding 100 μg m$^{-3}$.

We have added this statistics in Lines 373 to 375, Page 14 and in Lines 628 to 629, Page 23.

[Figure]

ReFig1. Time series of the hourly PM$_{2.5}$ obtained from observations (red circle), the ensemble mean of the analysis (blue line) and the ensemble mean of the background (the ensemble mean of the background, black line) in Beijing.

**12: Line 281, How do you decide the ensemble member of 50?**

We use the same EnSRF following Schwartz et al. (2012), in which the methodology/framework is similar to Whitaker and Hamill (2002). Whitaker and Hamill (2002) indicated the ensemble-mean RMS error is a function of ensemble size. When the ensemble size is larger than 50, the ensemble mean error is close to 0.19. So in this work, 50-member ensemble was chosen, following Schwartz et al. (2012) and Whitaker and Hamill (2002).

We have added some explanations in in Lines 247 to 248, Page 7.

**13: Line 349, Could you add mean distribution of PM2.5 concentration from the control and assimilation simulations in Figure 4? These will make the reader to understand a priori distribution and the adjustment of PM2.5 concentrations easily. Plotting mean observed PM2.5 concentrations on these maps will be even better.**

We added the spatial distribution of the PM$_{2.5}$ mass of the observations, the simulation of the control run, the analysis of expJ and expC, and also increments of

expJ and expC. The figure of the PM$_{2.5}$ mass differences was removed to save space. It is very clear that the analysis of expJ and that of expC are much different from the simulation of the control run.

Then we rewrote paragraph 2 in Section 5.2 in Lines 517, Page 19 to Lines 533, Page 20.

**14: Line 349, We can find adjustments over the SE Asia and India where you have no PM2.5 observation.**

The analysis increments (i.e. $\bar{x}^a - \bar{x}^b$) indicate the direct impact of assimilating PM$_{2.5}$ observation. They are determined by both the observation increments and the relative magnitudes of the forecast error and the observation error. From Figure 4 (e) and (f), we can see the increments of both assimilation experiments are distributed around the locations of observations as expected. However, the impact of assimilating PM$_{2.5}$ observations is not limited to the areas where observations were located, observations information is also transported to other areas through the WRF-Chem forecast. Besides, the ensemble forecasts also partly contributed to the PM$_{2.5}$ mass differences (assimilation minus control). Therefore, the spatial distributions of the PM$_{2.5}$ mass in both assimilation experiments were significantly different from the control run. Thus we can find adjustments over the SE Asia and India where no PM2.5 observation is available.

We have added the above explanations in Lines 517 to 528, Page 19.

**15: Figure 5, Overlaying of a priori emissions (it will be flat lines) in Figure 5 may emphasize that the assimilation can generate the temporal variations in the emissions.**

I have overlaid a priori emissions (the dash dot line) in Figure 5.

**16: Line 375, Is the burning of crop residues limited in the JJJ region? Li et al. shows that the northern part of YRD also has large emissions from the burning.**

We are not sure. In expJ, some larger values for the optimized $E^a_{PM2.5}$ were also

obtained in the northern part of YRD region from 0000 UTC to 0015 UTC of 14 October and 15 October (see ReFig2). However, they were much smaller than that in JJJ. In addition, according to the Weekly Crop Residue Burning Monitoring Report traced by Environmental Satellite (data from the satellite Environment Center, Ministry of Environmental Protection), there were only 9 crop residue burning spots in Anhui province from 5 to 18 October 2014 and no crop residue burning spots were reported in YRD. Thus, we did not mention the burning of crop residues in YRD.

[Figure]

ReFig2 Spatial distribution of the mean differences between the ensemble mean analysis and the prior emissions of the unspeciated primary sources of $PM_{2.5}$ at the lowest model level

---

## Author Comment (AC2) · 2 Dec 2016

**Response to Reviewer #2's comments:**

We thank Referee # 2 for their thoughtful comments and suggestions that have helped to improve this manuscript. Our responses to comments (in bold style) and the corresponding changes to the manuscript are detailed below. In particular, according to the reviewer's suggestions, we have added two more simulations; we also substantially rewrote the texts in both the major context and summary section to emphasize reviewer's questions.

**General Comments:**

**The authors introduce a DA system based on an ensemble square root filter combined with WRF-Chem that assimilates surface observations of PM2.5 across China. The novelty is that they use both aerosol concentrations and emissions in their DA state vector (although it should be noted they did something very similar for CO2 in Peng et al ACP 2015).**

The method in this work is very similar to that used by Peters et al. (2007) and Peng et al. (2015) for $CO_2$ emission inversion, but it is still of novelty for applications in aerosol anthropogenic emissions. In Peters et al. (2007), $\lambda_{i,t}^{p}$ were all 1. And only natural $CO_2$ emissions (i.e., biospheric and oceanic emissions) were assimilated at the ecological scale due to the 'signal-to-noise' problem. Thus, the uncertainty of anthropogenic and other $CO_2$ emissions were ignored. Besides, the framework is more advanced compared to our previous work. In Peng et al. (2015), in order to generate $\lambda_{i,t}^{p}$, a set of ensemble forecasts were performed from time $t$ to $t+1$ to produce the $CO_2$ concentration fields, forced by the prescribed net $CO_2$ surface fluxes using the previous assimilated concentration fields as initial conditions. That means that the ensemble forecast were performed twice in that DA system and it was time consuming. However, in order to save computing time, we used the chemical fields $C_{i,t}^{f}$ available in the previous assimilation cycle to calculate $\lambda_{i,t}^{p}$ in this work. Thus, WRF-Chem runs to forecast only once during a DA cycle.

We have added the above paragraph in Lines 187, Page 7 to Lines 200, Page 8.

**While the main idea is interesting and the topic is certainly relevant to ACP, I recommend against publication for the following reasons: 1) no independent observations are used to evaluate results. While this is ok for the evaluation of forecasts, this is not good practice for the evaluation of analyses; 2) no proof is offered for the central contention that analyzing emissions *together with* concentrations improves results; 3) no proof is offered for the second central contention that this system improves emissions; 4) many assumptions are merely stated without due reference, deliberation or any kind of sensitivity study; 5) several conclusions are drawn based on irrelevant data (see my comments).**

**It should be noted that reviewer 1 mentions the first two points as well but is apparently more lenient.**

**Point 3 I find particularly important as this is a contention made by other authors as well (Tang et al, Miyazaki et al) with little in the form of proof. Models have errors, and analyzing emissions may simply balance out some of these errors without improving the emissions. Note that we do not have observations to evaluate those emissions but this cannot be used as an argument to forego proper scientific reasoning.**

**In addition I find the structure of the paper illogical, and missed important information on details of their DA system and several references to previous attempts at emission estimation. I hope the authors will continue this work but put more effort in stating their case convincingly, for this research topic is certainly worthwhile. Maybe my comments can be of some help towards improving this manuscript.**

Thanks for those comments which did help improving this manuscript. Please see the point-to-point answers as below.

1) We have used the independent observations to evaluate both the analyses and the forecasts. Please see the details in the revised manuscript (Lines 354 to 355, Page 13; Lines 502 to 515, Page 19; Lines 632, Page 23 to Lines 691, Page25).

2) An experiment of pure assimilation chemical ICs and the corresponding 48-h forecasts experiment were also performed for comparisons in the revised manuscript. Please see the details in the revised manuscript (Lines 432 to 434, Page 16; Lines 448 to 452, Page 17; Lines 513 to 533, Page 19; Lines 620 to 622, Page 23; Lines 665 to 704, Page 25).

3) The analyzed emissions are only the results of a mathematical optimum by utilizing observations. They are influenced greatly by the model errors and the observation errors. In addition, only surface $PM_{2.5}$ observations were applied in this work, which may lack abundant constraint on the sources of the secondary aerosol precursors. Moreover, we do not have direct or exact emission information to evaluate the analyzed emissions, which was a challenging to many emission inversion research teams (e.g. Tang et al, 2011; Miyazaki et al., 2012; Ding et al., 2015; Mclinden et al., 2016; etc.). Different from the situations that standard national emission inventories are reported by government as in USA, European or other countries, the rapid economic development and complexity of emission sources in China lead to large uncertainties in the current public available emission inventories. Thus it's impossible for us to conduct the direct evaluation on emissions. For this reason, we weaken our judgment in the text.

Nevertheless, our system considering the emission assimilation provided better simulation results and the improvement of emissions can be verified in terms of two aspects, the diurnal variation and the location of increased emissions. The diurnal variation in the assimilated emissions can be used to verify our judgment to some extent. Especially in the PRD and YRD, $E^a_{PM2.5}$ in the daytime were always larger than those in the night, which agreed well with Olivier et al. (2003), the WRAP (2006) and Wang et al. (2010). In addition, the locations of the larger values for the optimized $E^a_{PM2.5}$ in the JJJ region were in good agreement with the places of the crop residues burning traced by the environmental satellite of China. There were 10, 231, 37 and 3 crop residue burning spots in Hebei, Henan, Shandong and Shanxi province respectively from 5 to 11 October 2014 and the numbers are 7, 20, 5 and 21 respectively from 12 to 18 October 2014 (Weekly Crop Residue Burning Monitoring

Report traced by Environmental Satellite, 2015a, 2015b ).

We have added the above paragraph in Lines 588, Page 21 to Line 613, Page 22.

4) and 5), we have revised the manuscript according to the reviewer's suggestions.

**Abstract**

**1.    P 1, L 13: "The forecast model of emission scaling factors was developed by associating the time smoothing operator with WRF-Chem forecast chemical concentrations". Please rephrase, this sentence is hard to understand without reading the paper first.**

This sentence has been rephrased as: "The forecast model of emission scaling factors was developed by using the ensemble concentration ratios of the WRF-Chem forecast chemical concentrations and also the time smoothing operator".

We have rephrased these references in Lines 14 to 16, Page 1.

**Introduction**

**2. P 2, L 40: The authors seem unaware of a lot of previous work on ensemblebased DA: Sekiyama et al ACP 2010, Schutgens et al. ACP 2010a, Schutgens et al ACP 2010b. , Dai et al, *Env. Pol*. 2014, Rubin et al. ACP 2016, , Yumimoto et al GRL 2016. Please include those references.**

We have added these references in Lines 46 to 48, Page 2.

**3. P 2, L 50: Again, several references seem to be missing i.c. emission estimation. For aerosol: Zhang et al JGR 2005, Sekiyama et al. ACP 2010, Huneeus et al ACP 2012, Schutgens et al. Rem Sens 2012, Huneeus et al ACP 2013**

We have added these references in Lines 56 to 58, Page 3.

**Methodology**

4. **P 3, L 78: Please introduce the ENSRF in context of some other EnKF (EAKF, LEKF, LETKF). What is the reason for this choice of EnKF, what is it main strength/weakness?**

There are different versions of EnKF. The traditional EnKF with perturbed observations (Evensen 1994) introduces sampling errors by perturbing the observations. In contrast to the traditional EnKF, the EnSRF (Whitaker and Hamill, 2002) and the Ensemble Adjustment Kalman Filter (EAKF, developed by Anderson, 2001) obviate the need to perturb the observations. The local ensemble Kalman filtering (LEKF), a kind of EnSRF, was presented by Ott et al. (2002, 2004). It was computationally more efficient compared to the traditional EnKF, since it simultaneously assimilates the observations within a spatially local volume independently. The local Ensemble Transform Kalman Filter (LETKF, Hunt, 2007) integrates the advantages of the Ensemble Transform Kalman Filter (ETKF, developed by Bishop et al., 2001) and the LEKF. The computational cost of LETKF is much lower than that of the original LEKF because the former does not require an orthogonal basis. Though LETKF has more advantages, we still chose the same EnSRF as Schwartz et al. (2014) because we did not need to extend it to analyzing aerosol ICs, very similar to Schwartz et al. (2014).

We have added the above paragraph in Lines 205 to 219, Page 8.

5. **P 54 L 94: Change "can be approximated" to "will be approximated". It is by no means certain that this is a good approximation. Part of the evaluation & tuning of an EnKF involves exactly the sampling errors introduced by Eq 5 & 6**

We have changed this sentence in Line 235, Page 8.

6. **P 3: Since the DA depends on the forecast model's details, I suggest to first discuss the forecast model (and introduce C and λ, and only then the ENSRF)**

We have changed the orders of Section 2.1 and 2.2.

**7. P 4, L 105: Please provide a bit more information on the base setup of the model: domain size, grid resolution, major aerosol species**

We have added more information of the base set up of the model in Lines 101, Page 4 to Lines 114, Page 5.

**8. P 4, L 106: "to forecast the emission scaling factors and the aerosol control variables". What are the control variables? I guess the authors mean aerosol concentrations, please change this. Note that both C and λ form the state vector.**

We have revised this sentence in Line 88, Page 4.

**9. P 5, L 123: "for the lowest eight vertical levels": so the emission inventory included heights at which the emissions were injected? These heights are all within the boundary layer? Why are only the lowest 8 layers considered?**

In this work, the lowest 12 vertical levels were at ~ 12 m, 48 m, 98 m, 156 m, 232 m, 300 m, 400 m, 500 m, 600 m, 700 m, 850 m, and 1000 m respectively. So the lowest 12 layers were all within the boundary layer. And the lowest 8 layers were under 500 m.

The emission inventory did not include emission heights at which the emissions were injected, which may cause large uncertainties for model forecast. We prepared the prescribed emissions just following others research (Woo et al., 2003; de meij et al., 2006; Wang et al., 2010): the power generator emissions were interpolated for the lowest eight vertical levels. And other anthropogenic emissions were assigned totally to the 1$^{st}$ level.

Emissions are very small above 500 m for all pollutants. So only the lowest 8 layers are considered.

We have added more discussions about the prescribed emissions in Lines 112 to 114, Page 5; in Lines 117 to 120, Page 5.

**10. P 6, L 139: "$\kappa_{i,t}$ are random". I wouldn't call them random. I realize they are distributed around the mean $\overline{\kappa_t}$, , but they were calculated through a short-term forecast of WRF-Chem.**

Yes. The ensemble concentration ratio ($\kappa_{i,t}$) are distributed around the ensemble mean ($\overline{\kappa_t}$). And $\overline{\kappa_t} = \frac{1}{N}\sum_{i=1}^{N}\kappa_{i,t} = \frac{1}{N}\sum_{i=1}^{N}C_{i,t}^f/\overline{C_t^f} = \frac{1}{N*\overline{C_t^f}}\sum_{i=1}^{N}C_{i,t}^f = \overline{C_t^f}/\overline{C_t^f} = 1$. So they are actually distributed around 1.

We have removed random variables and changed this sentence as: 'so $\kappa_{i,t}$ are numbers distributed 1 and with ensemble mean values of 1'' in Line142, Page 6.

**11. P 6, L 144: "$\beta = 1.5$ was chosen in this study": This sounds like an arbitrary choice? Normally $\beta$ results from tuning a DA but no such exercise was done?**

Peters et al (2007) first used the time smooth operator to evaluate the $CO_2$ fluxed scaling factors: $\lambda_{i,t}^f = (\lambda_{i,t-2}^a + \lambda_{i,t-1}^a + \lambda_{i,t}^p)/3$ (P. 8, the last paragraph in Peters et al. 2007. Here, we use the same notation in our manuscript). In that work, $\lambda_{i,t}^p$ were all 1 (P. 11, below S3.3). The time smooth operator was very useful because $\lambda_{i,t}^f$ could gain useful information achieved by previous DA cycle through the using of $\lambda_{i,t-2}^a$ and $\lambda_{i,t-1}^a$. However, they had to assimilate natural CO2 emissions (i.e., biospheric and oceanic) at the ecological scale due to the 'signal-to-noise' problem. Thus, the uncertainty of anthropogenic and other CO2 emissions were ignored.

We used the time smooth operator following Peters et al. (2007). In order to optimize all $CO_2$ fluxes as a whole at grid scale, we first used the ensemble concentration ratio ($\kappa_{i,t}$) to calculate the ensemble prior emission scaling factors $\lambda_{i,t}^p$ in Peng et al. (2015). $\lambda_{i,t}^p$ were artificial data to generate the ensemble emissions. It was difficult to give the ensemble members of $\lambda_{i,t}^p$ for the ensemble-based emission inversion system. Perhaps it was the simplest way to generate this data at every assimilation cycle by directly using the standard normal distribution function. But the

inversion system failed to optimize the prior fluxes at grid scale due to the 'signal-to-noise' problem (We have done the experiment for $CO_2$ inversion). From the other aspect, if following Peters et al. (2007) completely, the time smooth operator was applied and $\lambda_{i,t}^{p} = 1$ was chosen. However, the scaling factors should be perturbed at the first assimilation cycle to generate the ensemble factors. Consequently, this inversion system failed to optimize the prior fluxes at grid scale due to the same 'signal-to-noise' problem (Peng et al., 2015). So other ways should be found to generate $\lambda_{i,t}^{p}$. In Peng et al., $\kappa_{i,t}$ was used to calculate $\lambda_{i,t}^{p}$, and it seemed effective.

In Peng et al. (2015), the ensemble spread of $\kappa_{i,t}$ was very small (ranging from 0 to 0.08 in most area at model-level 1), though the values of the ensemble spread of $\mathbf{C}_{i,t}^{f}$ after inflation could reach 1 to 14 ppmv in most area at model-level 1. Therefore, covariance inflation was used to keep it at a certain level. After covariance inflation, the ensemble spread of $\lambda_{i,t}^{a}$ ranged from 0.1 to 0.8 in most model area for $\beta = 70$. Besides, several sensitive experiments were performed to investigate $\beta$ (10, 50, 60, 70, 75, 80, 100). The ensemble spread of $\lambda_{i,t}^{a}$ ranged from 0.05 to 1.2 for $\beta =60$, 70, 75, 80. And the $CO_2$ DA system worked comparatively well for $\beta =60$, 70, 75, 80. Though $CO_2$ fluxes inversion was another topic, we mentioned it here because this experience was very helpful for us to develop the joint DA system for aerosol.

As for the $PM_{2.5}$ assimilation, we have done several sensitive experiments to determine the value of $\beta$ (1.2, 1.5, 1.8, 2, 2.5) by using $PM_{2.5}$ measurements at the five U.S. Embassies stations in China (We did not gain the $PM_{2.5}$ observations from the Ministry of Environmental Protection of China at that time, in August 2015). It showed that the DA system worked comparatively well for $\beta = 1.2$, 1.5 and 1.8. For these cases, the ensemble spread of $\lambda_{PM2.5}^{f}$ ranged from 0.1 to 1.25 in most model area. Thus, $\beta = 1.5$ was chosen for latter experiments.

The magnitudes of the ensemble spread of the emission scaling factors were very stable with time. For the joint DA experiment in this manuscript, the ensemble spread

of $\lambda_{PM2.5}^f$ ranged from 0.25 to 1 in most model area except India where we were not interested in and no observations were available (see details in ReFig. 1). In the manuscript, hourly area-averaged time series of the ensemble spread for $\lambda_{PM2.5}^f$ over JJJ, YRD, PRD were added in Figure 3d.

It is noted that there were very few negative values for $(\kappa_{i,t})_{inf}$ after inflation in some cases. A quality control procedure should be performed for $(\kappa_{i,t})_{inf}$ before further appliance: All these negative data were set as 0.001. Then $(\kappa_{i,t})_{inf}$ were re-centered to ensure the ensemble mean value of $(\kappa_{i,t})_{inf}$ were 1. We added this explanation in Lines 146 to 151, Page 6; Lines 489 to 496, Page 18.

[Figure]

ReFig. 1. Spatial distribution of the ensemble spread for $\lambda_{PM2.5}^f$ at the lowest model level at (a) 0000 UTC 6 October 2014; (b) 0000 UTC 7 October 2014; (c) 0000 UTC 8 October 2014; (d) 0000 UTC 9 October 2014 for $\beta = 1.5$;

**12. P 6, L 145: "As the concentrations were closely related to the emissions":
if I assume this refers to emissions and concentrations in the same grid-box
(given the mathematics of their DA system), this is a bold statement and needs
some strong arguments. I can see that during the dust season, Beijing area will
be heavily impacted by dust from Eastern China, invalidating your assumptions.
Even for pollution emissions, transport may actually be very important.**

It is true that transport is very important for aerosol or other air pollution. We
corrected the text as "**As the concentrations were closely related to the emissions
both locally and in the upwind regions.**"

As stated in Q11 in detail, the prior emission scaling factors $\lambda_{i,t}^{\mathrm{p}}$ were artificial
data to generate the ensemble emissions. We chose $\lambda_{i,t}^{\mathrm{p}} = (\kappa_{i,t})_{\mathrm{inf}}$ (4) (original Eq. 9)
only as a last resort. Though the concentrations are related to the emissions according
to the mass conservation equation, Eq. (4) is not strongly supported. However, same
as $(\kappa_{i,t})_{\mathrm{inf}}$, $\lambda_{i,t}^{\mathrm{p}}$ are numbers distributed around 1. From the perspective of
generating the ensemble emissions, $\lambda_{i,t}^{\mathrm{p}}$ can play the same role as other data, such as
the random numbers created by using the standard normal distribution function.
However, there are correlations among the grid-points of $(\kappa_{i,t})_{\mathrm{inf}}$ because $(\kappa_{i,t})_{\mathrm{inf}}$
are calculated through a short-term forecast of WRF-Chem. Thus, $\lambda_{i,t}^{\mathrm{p}}$ have the same
correlations as $(\kappa_{i,t})_{\mathrm{inf}}$. While the random numbers are totally different. There are no
correlations unless they are generated under certain correlations.

It is noted that the correlations among the grid-points of the prior emissions
depend on $\lambda_{i,t}^{\mathrm{p}}$. Maybe these correlations deviate far from the truth. However, the
correlations among the grid-points of the forecast emissions maybe come close to the
truth due to the appliance of the smooth operator after multiple iterations.

We have revised the sentence "**As the concentrations were closely related to
the emissions both locally and in the upwind regions**" in Lines 152 to 153, Page 6
and added the content of the above paragraph in Lines 157, Page 6 to Lines 165, Page
7.

**13. P 6, L 147: "concentration ratios $(\kappa_{i,t})_{\text{inf}}$ served as the prior emission scaling factors $\lambda_{i,t}^{\text{p}}$." So the concentrations themselves were not inflated, as is usually done in EnKF? What is the justification for this? Shouldn't the scaling factors be perturbed according to the uncertainty in emission inventories and parametrizations?**

Posterior multiplicative inflation was applied for only the concentration analysis aiming to maintain ensemble spread.

As for the emission scaling factors, posterior multiplicative inflation was not used. Besides, they are not perturbed according to the uncertainty in emission inventories and parametrizations. Since $\lambda_{i,t}^{\text{p}}$ are calculated through a short-term forecast of WRF-Chem, $\lambda_{i,t}^{\text{f}}$ have deterministic values from the time smooth operator.

We have addressed the posterior multiplicative inflation, plus the covariance localization, in Lines 247 to 256, Page 10.

**14. P 6, L 152: I suspect that Eq 10 is missing a factor 0.5. The prior and analysis scale factors are previous times are averaged.**

It is right that the prior and analysis scale factors of previous times are averaged, but a factor 0.5 is not missed. In Equation (5) (original Eq. 10), j starts from t-M+1. Thus, M times of scale factors (the prior and M-1 analysis scale factors) are used to calculate $\lambda_{i,t}^{\text{f}}$. For example, in our manuscript, M = 4. Thus, $\lambda_{i,t}^{\text{p}}$, $\lambda_{i,t-1}^{\text{a}}$, $\lambda_{i,t-2}^{\text{a}}$, and $\lambda_{i,t-3}^{\text{a}}$ are used. Therefore, the denominator in the right hand of Equation (5) is 1/4.

**15. P 6, L 153: Again, a rather arbitrary choice (M=4)? How does this relate to the DA cycle?**

According to the smooth operator, the ensemble mean values of $\lambda_{i,t}^{\text{f}}$ depend on the ensemble mean of $\lambda_{i,t-M+1}^{\text{a}}$, $\cdots$, $\lambda_{i,t-2}^{\text{a}}$, $\lambda_{i,t-1}^{\text{a}}$, $\lambda_{i,t}^{\text{p}}$, where the ensemble means of $\lambda_{i,t}^{\text{p}}$ are all 1. After multiple iterations, the smooth operator can give comparatively

good estimation for $\lambda_{i,t}^{\text{f}}$ since anthropogenic emissions are stable at a certain time scale (Mijling et al., 2012).

Peters et al (2007) chose M=3 ($\lambda_{i,t-2}^{\text{a}}$, $\lambda_{i,t-1}^{\text{a}}$ and $\lambda_{i,t}^{\text{p}}$ were used to calculate $\lambda_{i,t}^{\text{f}}$) for $CO_2$ fluxes inversion. They indicated that it was a compromise between prescribing prior $CO_2$ fluxes at each step and letting the system propagate all information from one step to the next without any guidance (in L 3, P 11). They also pointed out that the latter will work fine for the North American fluxes which were strongly constrained by observations. Similar to Peters et al. (2007), fewer states are used to calculate $\lambda_{i,t}^{\text{f}}$ for the joint DA system for aerosol in this manuscript.

In the revised manuscript, we have added some explanation in Lines 171 to 177, Page 7 and some results in Lines 539 to 541, Page 20.

**16. P 6, L 159: "emission inventories". Except in the case of dust, sea-salt etc. Or are these not perturbed? If not, why are they not perturbed (surely they are uncertain as well)? Actually, the authors are rather sparse in their information. Is each species perturbed independently from the others? What is the level of perturbation? Are neighbouring grid-points perturbed independently or do you assume correlations?**

In the assimilation part, we had applied 4 independent scaling factors: $\lambda_{\text{PM2.5}}$, $\lambda_{\text{SO2}}$, $\lambda_{\text{NO}}$ and $\lambda_{\text{NH3}}$. Both the forecast emissions (perturbed emissions) and the assimilated emissions were calculated according to EQ (6) : $\mathbf{E}_{i,t} = \lambda_{i,t}\mathbf{E}_t^{\text{p}}$ (original Eq. 11). $\lambda_{\text{PM2.5}}$ were used to calculated $\mathbf{E}_{\text{PM2.5i}}$, $\mathbf{E}_{\text{PM2.5j}}$, $\mathbf{E}_{\text{SO4i}}$, $\mathbf{E}_{\text{SO4j}}$, $\mathbf{E}_{\text{NO3i}}$, and $\mathbf{E}_{\text{NO3j}}$ (see details in 2.3.1). $\lambda_{\text{SO2}}$, $\lambda_{\text{NO}}$ and $\lambda_{\text{NH3}}$ were used to calculate $\mathbf{E}_{\text{SO2}}$, $\mathbf{E}_{\text{NO}}$ and $\mathbf{E}_{\text{NH3}}$. In this study, only the species of the emission inventories mentioned above were perturbed (or updated according to the assimilated scaling factors).

Other inorganic species of the anthropogenic emission, such as $\mathbf{E}_{\text{EC}}$ and $\mathbf{E}_{\text{ORG}}$, are not perturbed for WRF-Chem, which is a limitation of this manuscript. However, other anthropogenic emissions, such as $\mathbf{E}_{\text{PM2.5}}$, $\mathbf{E}_{\text{SO4}}$ and $\mathbf{E}_{\text{NO3}}$ are much larger

than $\mathbf{E}_{EC}$ and $\mathbf{E}_{ORG}$ in most area of China, and the ensemble spreads of the aerosol concentrations largely depend on the uncertainties of those anthropogenic emissions. Besides, model errors arisen from the meteorology, the emissions and the chemical model itself are compensated to some extent through the use of multiplicative inflation. In other words, the ensemble spread of the concentrations can be kept at a certain level though $\mathbf{E}_{EC}$ and $\mathbf{E}_{ORG}$, are not perturbed.

Natural emissions, such as dust and sea salt were not perturbed explicitly when the forecast emissions were generated. However, emissions of dust and sea salt were parameterized within the GOCART model (Chin et al., 2002). Within the DA system, varying meteorology across the members implicitly perturbed dust and sea salt emissions.

We have added the above two paragraphs in Lines 320, Page 12 to Lines 334, Page 13.

No other perturbations are added to the scaling factors. And no other correlations are assumed for the scaling factors. As stated above, both the forecast emissions (perturbed emissions) and the assimilated emissions were calculated according to EQ (6) : $\mathbf{E}_{i,t} = \boldsymbol{\lambda}_{i,t}\mathbf{E}_t^{\mathrm{p}}$ (original Eq. 11). The correlations among the grid-points of the forecast emissions depend on the correlations among the grid-points of $\boldsymbol{\lambda}_{i,t}^{\mathrm{f}}$. See some detail in Q.12 and in Line 182 to 186, Page 7.

**17. P 7, L 175: "the state variables of the analysis of the ICs were the 15 WRFChem/ GOCART aerosol variables." This should have been mentioned earlier, maybe line 101.**

We have moved this to lines 242 to 244, page 9.

**18. P 7, L 184: "($\lambda$PM2.5, $\lambda$SO2, $\lambda$NO and $\lambda$NH3)" This line and the following paragraph suggest that the authors keep the $\mathbf{E}_{EC}$ and $\mathbf{E}_{ORG}$ constant? They do not matter? I rather think they do. By the way, this paragraph might be**

**rewritten to improve readability.**

Yes, we keep the $\mathbf{E}_{EC}$ and $\mathbf{E}_{ORG}$ constant during the joint DA experiment, which is a limitation in this manuscript. It is true that these emissions are also important for the atmosphere aerosol. The reason we did not assimilate $\mathbf{E}_{EC}$, $\mathbf{E}_{ORG}$ is that only the $PM_{2.5}$ measurements are used in this DA experiment. However, the sources of the aerosols (especially organic aerosols) are so complex that our knowledge of their formation mechanisms is far from clear. Though it is technically possible to have all emissions assimilated, with such limited observations adding more control variables would cause much more uncertainties in the system which might lead to unreasonable analysis. This is our first attempt to simultaneously optimize the chemical ICs and emission input. In future work, when gas-phase observations of $SO_2$, $NO_2$ and $O_3$ are used and more aerosol species observations are available, perhaps more emissions are assimilated, similar to Tang et al. (2011).

We have added the above paragraph in Lines 300 to 308, Page 12.

We have also rewritten this paragraph in Lines 268 to 276, Page 10.

**19. P 8, L 208: The authors never explain how the system is started up. Some initial perturbation in concentrations and/or emissions must be assumed.**

We have rewritten some part of in Sec. 4.2 in Lines 424 to 431, Page 16.

**20. P 9, L 247: "$\varepsilon_r = r\varepsilon_0\sqrt{\Delta x/L}$," Can the authors provide a reference for this form of the representation error? Why do they choose L=3 km? How can it be that the representation error is a function of the measurement error? These are two independent error sources.**

We calculated the representation errors completely following Schwartz et al. (2012), who followed Elbern et al. (2007) and Pagowski et al. (2010). Elbern et al. (2007) developed this scheme firstly based on the research of the European organizations. In Elbern et al. (2007), L= 20, 10, 4, 2, 1 and 3 km for Remote, Rural, Suburban, Urban, Traffic and Unknown station type (P 3758) respectively. We had

added some information of the scheme in Lines 366 to 367, Page 13.

**21. P 9, L 252-255: Some statistics on how often this happened would be appreciated.**

The numbers of the observations were about 17700. Among them 8 observations were discarded because they were larger than 800 $\mu g \ m^{-3}$ and 243 (around 1.5%) were discarded due to the ensemble mean of the first guess departure exceeding 100 $\mu g \ m^{-3}$.

We added this statistics in Lines 373 to 375, Page 13.

**22. P 10, L 261: "The horizontal grid spacing was 40.5 km and there were 262 57 vertical levels with the model top at 10 hPa." This sort of information should be in Sect 2.2.1**

We have moved this sentence in Sect. 2.1.1.

**23. P 1, L 265: "initialization and spin-up procedures" Please briefly state the spinup procedure. For how long was the ensemble run before the first DA happened?**

We have done initialization experiments from 0000 UTC 1 October to 2300 UTC 4 October 2014. And we have rewritten the last paragraph in Sect. 4.1 in Lines 413 to 416, Page 15.

**24. P11, L 279: "clean oceanic conditions." Does this mean that over land you assumed seasalt aerosol as LBC?**

Actually the LBCs for chemistry/aerosol fields were idealized profiles embedded within the WRF/Chem model. It's not only for the clean oceanic conditions. We have corrected the text. The differences between the idealized profile and real boundary conditions may bring some errors for the boundary region but since our focus is centered in the JJJ, YRD and PRD regions that far from the boundary region. The impacts would be negligible.

We have corrected the text in Lines 398 to 399, Page 14.

**25. P 11, L 280: "standard Gaussian random noise". Please briefly state what standard deviations you assumed, and how you dealt with negative emissions.**

We perturbed the anthropogenic emissions following Schwartz et al. (2012).

For possible negative perturbed emissions, they were set as $E_{ip}^*(\eta, t) = 0.001 *$ $E_p(\eta, t)$. This will increase the prescribed emissions more or less. However, only very few data were negative. So, this influence can be negligible.

It should be noted that the perturbed emissions were only used in the spin-up procedure and expC.

We have rewritten this part in Lines 403 to 412, Page 16.

**26. P 13, L 336: "These statistics were calculated against observations over all the analyses" If I understand the authors, the same observations that were assimilated are here used to evaluate the results. This likely explains the high correlations. The authors should make it clear this is not an independent evaluation but merely a sanity check.**

We have added the independent observations to evaluate the analysis in Lines 501 to 515, Page 19.

**27. P 13, L 356: "These results indicate that DA greatly improved the ICs." This is rather bold as you have not used independent observations to evaluate the ICs. Obviously, if you nudge the model towards observations, the model will do better. Please remove this sentence.**

We have used the independent observations to evaluate the analysis. We also removed this sentence.

**28. P 13, L 363: "the optimized PM2.5 scaling factor, λPM2.5a, showed an**

obvious variation with time, as did the optimized unspeciated primary sources of PM2.5, EPM2.5a" From the authors explanation of how their system works, I do not understand why λPM2.5 and EPM2.5 would have a different (if only slightly) time evolution. Is this because they are regional averages?

Thanks for pointing out this error! The $\lambda_{PM2.5}^{f}$, $\lambda_{SO2}^{f}$, $\lambda_{NO}^{f}$ and $\lambda_{NH3}^{f}$ were 1 hour earlier than the $\mathbf{E}_{PM2.5}^{f}$, $\mathbf{E}_{SO2}^{f}$, $\mathbf{E}_{NO}^{f}$ and $\mathbf{E}_{NH3}^{f}$ in the original plot as I made a mistake when extracting those values.

ReFig. 2 (also updated in the manuscript) shows the right results. It shows that the $\mathbf{E}_{PM2.5}^{a}$ change along with $\lambda_{PM2.5}^{a}$.

[Figure]

ReFig. 2. Hourly area-averaged time series of emission scaling factors (black) extracted from the ensemble mean of the analyzed $\lambda_{PM2.5}^{a}$ and the corresponding analyzed unspeciated primary PM$_{2.5}$ emissions $\mathbf{E}_{PM2.5}^{a}$ (blue) over the three sub-regions: (a) Beijing–Tianjin–Hebei region; (b) Yangtze River delta; and (c) Pearl River delta.

**29. P 13, L 379: "as the system is optimized based on ambient concentrations in which the transport and transformation processes are not directly taken into account" But surely transport is important? Maybe a Kalman smoother would have been a better system to solve this problem.**

We think transport is as important as transformation. In our DA experiments, the

PM$_{2.5}$ measurements network was still spatially sparse and heterogeneous. Almost all measurements were located in the city and no data available in the rural region. However, the crop residues burning always occur in rural region. So the PM$_{2.5}$ measurements network can only capture the burning information a few hours later. It is right that a Kalman smoother would have been a better system to solve this problem.

We have added some explanation in Lines 557 to 565, Page 20.

**30. P 14, L 388: "at the lowest model level" Why do you only discuss emissions at lowest level? Are they much larger than those at higher levels? Surely it is the vertically integrated emissions that is important for the amount of particulate matter entering the atmosphere?**

Yes, the emissions at lowest level were much larger than those at higher levels. So the time-averaged differences between the ensemble mean analysis and the prior values of the unspeciated primary sources of PM$_{2.5}$ at higher levels were negligible (See ReFig. 3). Thus we only discussed emissions at lowest level.

We have added some explanation in Lines 572 to 574, Page 21.

[Figure]

ReFig. 3 Spatial distribution of (a) the prior unspeciated primary sources of PM$_{2.5}$ ($\mu$g m$^{-2}$ s$^{-1}$) and (b) the time-averaged differences between the ensemble mean analysis and the prior values ($\mu$g $\cdot$m$^{-2}$ s$^{-1}$) of the vertically integrated emissions from level 2 to level 8 averaged

over all hours from 6 to 16 October 2014.

**31. P 15, L 406: "Our assimilated PM2.5 and NOx emissions were in good agreement with this trend". The DA experiments reported here cover a period of a few weeks, so how can you compare that to a trend over 15 years?**

This conclusion was really arbitrary. We have removed related sentences.

**32. P 17, L 470: "However, these results are still better than those obtained with the pure adjustment of ICs that lead to improvements in the first 12-h forecasts (Jiang et al., 2013; Schwartz et al., 2014)." This conclusion is baseless as Jiang et al use a different DA system (3D-VAR) with different observations (PM10) and Schwartz et al use a different domain (USA).**

In the revised manuscript, the experiment of pure assimilation chemical ICs and the corresponding 48-h forecasts experiment were also performed for comparison. It seemed that the forecasts with the joint adjustment were always much better than the forecasts with only the optimized ICs for almost all the forecasts in the PRD and YRD. Please see the details in the manuscript (Lines 432 to 434, Page 16; Lines 448 to 452, Page 17; Lines 513 to 533, Page 19; Lines 620 to 622, Page 23; Lines 665 to 704, Page 25).

**33. Figure 1: What is F? How is it related to Eq 1?**

It was **E**. We have corrected it in Figure 1.

---

## Author Response (AR2)

Feb. 7, 2017.

*Atmos. Chem. Phys.*

RE: Manuscript Number: acp-2016-732

Dear Editors:

Thank you very much for your kind decision letter on our paper entitled "**Improving**

**PM$_{2.5}$ forecast over China by the joint adjustment of initial conditions and source**

**emissions with an ensemble Kalman**" (acp-2016-732). We are grateful for the helpful comments from you and the reviewers. We have changed the manuscript according to the reviewer's suggestions. We hope this manuscript will be published in

ACP. We are looking forward to hearing from you soon.

Sincerely Yours,

Zhen Peng

**Response to Reviewer #1's comments:**

We thank Reviewer # 1 for his thoughtful comments and suggestions that have helped to improve this manuscript. Our responses to comments (in bold style) and the corresponding changes to the manuscript are detailed below.

**The revised manuscript by Peng et al. is much approved, and I thank the authors for the efforts to address my review. I believe the revised paper is suitable for publication. Only two questions still in need of attention.**

**1.Table 1, lines 503-516,Could you mention that expC has better RMSE and CORR than expJ in JJJ and reasons for this?**

It is interesting to note that expC has better RMSE and CORR than expJ but poor bias in JJJ. And expC has better bias and RMSE than expJ but poor CORR in PRD. Maybe small number of samples caused the uncertainties of the statics. However, the differences were very small. The analysis of both experiments were very similar.

We have added some discussions in Lines 536-539

**2.Revised figure 4(c),I believe that both Figure 4 in the previous manuscript and Figure 4(e) in the revised manuscript shows PM2.5 mass difference (assimilation minus control) for expJ. However, distribution is different before and after the revise; negative increments over India and Southeast Asia are disappeared in the revised one.**

I am sorry that I have changed the figure in the revised manuscript. In ACPD, Figure 4 was assimilation minus control (see ReFig. 1). In the revised manuscript, Figure 4e was increment, assimilation minus background. They are not the same.

We have added some discussions in Lines 545-551 for the negative PM2.5 mass difference (assimilation minus control) over India and Southeast

[Figure]

ReFig. 1(same as figure 4 in ACPD. PM$_{2.5}$ mass differences (assimilation minus control, μg m$^{-3}$) at the lowest model level averaged over all hours from 6 to 16

October 2014.

**Response to Reviewer #3's comments:**

We thank Reviewer # 3 for their thoughtful comments and suggestions that have helped to improve this manuscript. Our responses to comments (in bold style) and the corresponding changes to the manuscript are detailed below.

**The authors present the results of a forecasting system that assimilates both initial hourly aerosol concentration and emission fluxes in order to improve the forecasting of particulate matter concentrations over China. To evaluate the performance of this system the forecasted concentrations are contrasted on one hand with independent observations not assimilated by the system and on the other hand against a control run without any assimilation and a forecast experiment only assimilating initial conditions but no emissions. The forecast is conduct for all China but a more in depth analysis is conducted in three regions experiencing stronger pollution levels. These three regions are the Beijing-Tianjin-Hebei region, the Yangtze River delta and the Pearl River Delta. The authors present results illustrating that the forecast assimilating initial conditions and emissions performs much better than the control simulation. Performance analysis in the three above-mentioned regions suggests that the system achieves improvements for almost all 48-h forecast in two of them while in the third one the improvement is more limited. Similarly the performance of the joint assimilation compared to the one only assimilating initial conditions shows improvement in two of the regions.**

**The results presented in the manuscript are interesting, however the authors conduct only a shallow analysis of the results and do not discuss how some of the assumptions made in the system affect the result. Although I recommend this paper for publication I would suggest the authors extend the discussion of the results addressing some of the topics highlighted below. When presenting a new inversion system, in addition of presenting the main results (if it works or not), the limitations of the system and their impact should also be presented.**

**General comments**

**The authors assume prior emissions constant in time but it is well known that emissions are not constant throughout the day. Why were emissions considered constant throughout the day and also throughout the week? How much of the improvement in performance of the system comes from this assumption? How much better does the control perform when variable emissions within the day are allowed? Furthermore, the implications of not perturbing emissions of elemental carbon and organic carbon should be included in the manuscript. How does this affect the forecast? How realistic is the result provided by the system with this constrain?**

It is true that emissions are not constant throughout the day. As also found in earlier modeling studies, the temporal allocation of emissions plays essential roles for chemical forecasts (de Meij et al., 2006; Wang et al, 2009). However, it still lacks resolution of temporal allocations at shorter but critical (e.g.,day-of-week, diurnal) scales. In order to keep objective for the prior anthropogenic emissions, no time variation was added in this work. However, vertical allocations of anthropogenic emissions are considered. The power generator emissions were interpolated for the lowest eight vertical levels (Woo et al., 2003; de meij et al., 2006; Wang et al., 2010). Other anthropogenic emissions were assigned totally to the $1^{st}$ level.

The constant anthropogenic emissions will worsen the chemical forecasts of the control run. Wang et al. (2010) pointed that surface $NO_2$ and $SO_2$ concentrations are reduced by respectively 3-7 and 6-12 ppbv over major cities and industrial areas when the emissions are allocated temporally and spatially. And surface $O_3$ concentrations are higher by 4-8 ppbv at night and 2-4 ppbv in daytime over broad areas of northern, eastern and central China. For the joint DA system itself, it cannot benefit from the constant prior anthropogenic emissions. But the normalized RMSE in Figure 10g decreased due to the poor forecasts of control run. The control run will perform better when variable emissions within the day are allowed, especially during the night. As a result, the relative reduction in RMSE could not be so large during the night.

113   The above discussions are added in Lines 761-770.

115   For the assimilation of $BC_1$, $BC_2$, $OC_1$ and $OC_2$, the difference between expC

116 and expJ can be seen as the perturbing emissions of $E_{EC}$ and $E_{ORG}$ since $E_{EC}$ and

117 $E_{ORG}$ of the anthropogenic emissions were not assimilated in expJ. ReFig. 1 and

118 ReFig. 2 show mass differences (assimilation minus control, $\mu g\ m^{-3}$) at the lowest

119 model level averaged over all hours from 6 to 16 October 2014 in expJ and expE

120 respectively for $BC_1$, $BC_2$, $OC_1$ and $OC_2$. Though we cannot conclude which one

121 is closer to the truth due to the lack of observations, $OC_1$ and $OC_2$ are changed

122 contributed to the $PM_{2.5}$ assimilation in both experiments, which suggests that the

123 influence of not perturbing $E_{EC}$ and $E_{ORG}$ could be negligible. The reason that the

124 differences of $BC_1$ and $BC_2$ are close to zero is that the magnitude of $BC_1$ and

125 $BC_2$ are too small.

126   The above discussions are added in Lines 775-781

[Figure]

ReFig. 1. (a) $BC_1$; (b) $BC_2$; (c) $OC_1$ and (d) $OC_2$ mass differences (assimilation minus control, μg m$^{-3}$) at the lowest model level averaged over all hours from 6 to 16

October 2014 in expJ.

[Figure]

ReFig. 2. (a) **BC$_1$** ; (b) **BC$_2$** ; (c) **OC$_1$** and (d) **OC$_2$**(a) **BC$_1$**; (b) **BC$_2$**; (c) **OC$_1$** and (d) **OC$_2$** mass differences (assimilation minus control, μg m$^{-3}$) at the lowest model level averaged over all hours from 6 to 16 October 2014 in expE.

**The authors examine first the performance of the system by comparing the analysis of both assimilation experiments (expC and expJ) to the observations and then the forecast. It is interesting to note that when the analysis of both experiments are examined a better performance is obtained in PRD and JJJ when only initial conditions (IC) are assimilated (i.e. expC). However, when comparing the forecasts between both experiments, expJ performs better than the forecast of expC. What are the implications of this result? Furthermore, the authors provide a too simplistic analysis of the performance of the forecast in the**

**147** **three regions. Yes it is true that expJ improves with respect to the control and**
**148** **expC in YRD and PRD, but this is mostly for daytime, during night-time the**
**149** **improvement is very similar in three regions. In YRD, the performance is**
**150** **actually deteriorated during nighttime and in JJJ there is either deterioration or**
**151** **no improvement after 24 hr forecast for both assimilation experiments. Although**
**152** **the authors suggest that this is mainly to a good performance of the model**
**153** **during nighttime, this is not enough I believe. Why is the performance of the**
**154** **control run better during night? Why does the assimilation have so little impact**
**155** **during night? Why should the model have better performance for nocturnal**
**156** **conditions? Was it tuned under such conditions? Do the a priori emissions**
**157** **provided, the ones considered constant, correspond to night emissions? I would**
**158** **suggest the authors spend a bit more trying to address this issue as they have**
**159** **done so far.**

**160** From Table 1, the biases were -10.3, -1.6 and 4.7µg $m^{-3}$ for JJJ, YRD and PRD,

**161** respectively, and RMSEs were 66.9, 15.3, 16.1µg $m^{-3}$ respectively in expJ. The biases

**162** were -12.2, -2.4 and -2.3µg $m^{-3}$ for JJJ, YRD and PRD, respectively, and RMSEs were

**163** 64, 17.3, 15.6µg $m^{-3}$ respectively, in expC. Thus, expC has better RMSE and CORR

**164** than expJ but poor bias in JJJ. And expC has better bias and RMSE than expJ but poor

**165** CORR in PRD. Maybe small number of samples caused the uncertainties of the statics.

**166** However, the differences were very small. The analysis of both experiments were

**167** very similar.

**168** . When comparing the forecasts between both experiments, expJ performed

**169** much better than the forecast of expC. This could be attributed much to the emissions

**170** since the ICs of both forecasts were very similar. In the forecast experiment of expC,

**171** the emissions were the default monthly anthropogenic emissions. While in the

**172** forecast experiment of expJ, the assimilated emissions were different much from the

**173** default monthly anthropogenic emissions (see Figure 5 and 6). Also, there was diurnal

**174** variation.

**175** The above discussions are added in Lines 536-539 and Lines 721-726

**176**

The improvements were comparatively small in PRD in the daytime. And the
performance was actually deteriorated in YRD during the same time. One of the
possible reasons was that chemical model performed sufficiently well during daytime
when the boundary layer was unstable and therefore the further improvement was
more difficult. And there were always large errors during the night when the boundary
layer was stable, so that large improvements could be obtained. The other possible
reason can be attributed to the a priori constant emissions. The differences between
the optimized $PM_{2.5}$ emissions and the prior emissions were comparatively small
during the day, but the optimized $PM_{2.5}$ emissions were much smaller than the a prior
emissions during the night. So that the control run could performed worse during the
night and it could performed well during the day. Given the a priori variable
emissions provided, the control run will perform better during the night.

We have added the above discussions in Lines 680-690

**If the difference between the control run and expC can be seen as the**
**contribution of assimilating concentrations, can the difference between expC and**
**expJ as the impact of assimilating emissions? If so, is it really worth if to**
**assimilate both? Why wasn't there and expE conducted where only emissions**
**were assimilated? Figure 8 suggests that in most of the days in the three cities,**
**the fact of assimilating only IC has little impact on the forecast. Figure 9 also**
**illustrates that most of the improvement comes when emissions are assimilated.**
**What if only emissions were to be assimilated, could that be enough? I suggest**
**the authors include a discussion section where this is addressed.**

The difference between the control run and expC can be seen as the contribution
of assimilating concentrations, and the difference between expC and expJ can be seen
as the impact of assimilating emissions. Though the fact of assimilating only IC has
little impact on the forecast in most of the days in the three cities (See Figure 9) and
most of the improvement comes when emissions are assimilated (See Figure 10), it
was still worth to simultaneously assimilate the chemical ICs and emission. We have
performed the expE for 7 days where only emissions were assimilated during our limited time. In order to remove the influence of the cumulative errors resulting from the initialization and spin-up experiment, the chemical ICs were first assimilated from

2000 UTC to 2300 UTC 4 October 2014. The first 50 ensemble chemical fields were drawn from the WRF-Chem ensemble forecasts valid at 2000 UTC 4 October 2014.

Then expE were performed from 0000 UTC 5 October 2014 to 0000 UTC 12 October

2014.

In expE, the chemical concentrations can be updated by the WRF-Chem model simulations with the assimilated emissions as the initial field in each DA cycle (see

ReFig. 3). That means that the 50-member ensemble forecasts were performed twice and it was time consuming.

On the other hand, it seemed that better concentration analysis could be obtained in expJ due to the simultaneous assimilation of ICs and emissions. Both the background $PM_{2.5}$ and the analysis $PM_{2.5}$ in the assimilation experiments were comparatively near to the observations (see ReFig. 4) in expJ. However, both the background and the analysis $PM_{2.5}$ deviated markedly from the observations (see

ReFig. 5) in expE. Especially in Beijing, the performance is deteriorated for $PM_{2.5}$

observations above 200 μg $m^{-3}$ when an intense pollution events occurred. This will lead to larger uncertainties for the emission inversion. Also the improvement of $PM_{2.5}$

forecasts will be limited due to the comparatively poor chemical ICs.

We added a **discussion in Lines 781-792**

[Figure]

ReFig. 3 Flow chart of the DA system of expE

[Figure]

ReFig. 4. Time series of the hourly PM$_{2.5}$ obtained from observations (circle), control
run (black line), analysis (red line), and background (green line) in three megacities:
(a) Beijing; (b) Shanghai; and (c) Guangzhou in expJ.

[Figure]

ReFig. 5. Time series of the hourly PM$_{2.5}$ obtained from observations (circle), control
run (black line), analysis (red line), and background (green line) in three megacities:
(a) Beijing; (b) Shanghai; and (c) Guangzhou in expE.

**The assimilations system needs further description. The authors describe
how the observation error covariance matrix (R) is defined but do not do the**

**same for the background error covariance matrix (Pb). How is Pb defined? The authors should explain this in the manuscript. Furthermore, observations from 77 stations were assimilated and observations from another set of 77 stations were used for verification purposes. However, in the three regions of interest in the manuscript; namely JJJ, YRD, and PRD, it is not clear how many stations were assimilated and how many were used in the verification. This number is provided in the caption of Figure 1 but should also be included in the text. Please also clarify if all these verification stations are used to compute the statistics presented in Figure 9.**

$\mathbf{P}^b$ is defined in Line 241.

We have added the numbers of the stations used for assimilation and those for verification in the three regions in lines 489-492.

All these verification stations are used to compute the statistics presented in Figure 10 and we have clarified.

**Specific comments**

**1 Lines 30-31: Acronyms should be defined.**

We have defined the acronyms in Lines 22-27

**2 Lines 79-81: Structure of the paper described is not consistent with actual structure of the paper. There are 6 sections in the manuscript and only 5 according to text in last sentence of section 1.**

Thanks for pointing out this mistake and we have revised in Lines 78-81.

**3 Line 131: Sub index i should be defined. It is clear from the text what it stands for but should be introduced anyway.**

We have defined it in Lines 131-134.

**4 Line 132: Why is it t-2 for the emissions and t-1 for the concentrations (line 131)? Is it a mistake and it should be t-1 for both? If not, please explain.**

Thanks for pointing out this error. I have corrected it in Line 133.

**5 Please explain which criteria was applied to define the limits (0.1 and 1.25) to**

**the spread of (Ki,t)inf. How were they defined?**

In Peng et al. (2015), several sensitive experiments were performed to investigate $\beta$

(10, 50, 60, 70, 75, 80, 100). The ensemble spread of $\lambda_{i,t}^a$ ranged from 0.05 to 1.25

for $\beta = 60$, 70, 75, 80. And the $CO_2$ DA system worked comparatively well for

$\beta = 60$, 70, 75, 80. For $\beta = 10$, 50, the impact of assimilation was small due to the small ensemble spread; For $\beta = 100$ the assimilated $CO_2$ fluxes deviated markedly from the "true" $CO_2$ fluxes due to too large ensemble spread. Therefore, in this work,

$\beta = 1.5$ was chosen to make ensure the ensemble spread of $(\kappa_{i,t})_{inf}$ ranged from

0.1 to 1.25 in this study.

We have added some explanations in Lines 148-152.

**6 Line 150: Why are the negative values set to 0.001 and not simply 0? Please**

**explain.**

We have no special reasons to set the negative values as 0.001. It is also fine to set them as 0. We have added this in Lines 157-158.

**7 Line 322: Remove "which is a limitation of this manuscript". It is already**

**stated in lines 300 and 301.**

We have removed this sentence.

**8 Lines 352-356: Explain the criteria used to select the stations that would be**

**used for verification and those used in the assimilation? How many of each are in**

**the different regions. The total number of stations in each region is provided but**

**it is not said how many of them are for validation or verification purposes.**

There are altogether 906 national control measurement sites over China. The reason we did not use all the measurements is that we also have done a sensitive experiment by using $PM_{2.5}$ measurements at the five U.S. Embassies stations and

PM$_{2.5}$ measurements at 34 monitoring sites in Beijing from the national

Environmental Monitoring Center except sensitive experiments by only using PM$_{2.5}$

measurements at the five U.S Embassies stations. The DA system could ingest the observations effectively by only using PM$_{2.5}$ measurements at the five U.S Embassies stations. We thought we might gain even better assimilation results in

Beijing-Tianjin-Hebei region for more assimilation measurements. However, it is unexpected that the impact of the ensemble assimilation on ICs and emissions were almost negligible and no improvements were gained for PM$_{2.5}$ forecast. So far we did not know the exact reason. But, the 34 sites in Beijing are fall into 8 model grids (Chen et al., 2016). So many observations are fall into one model grids. However,

Chemicals are influenced greatly by the local emissions. Observations are not in good agreement with each other though they are fall into the same model grid. Therefore, the observation error covariance matrix **R** may include much noise. And the ensemble data assimilation system at this stage could dot absorb useful observation information effectively. However, further investigations are needed to resolve the question. In this work, only a few measurements were assimilated for simplicity since the DA system performed well by only using PM$_{2.5}$ measurements at the five U.S

Embassies stations.

The PM$_{2.5}$ observation sites spanned most of central and eastern China but were primarily located in urban and suburban areas. So it always happened that there are more than one observation sites in certain city. We randomly selected one observation site in a city for assimilation experiment and one for verification purposes since we did not know the exact station type. Altogether 77 stations were selected for the PM$_{2.5}$

assimilation experiment and another 77 stations were selected for verification. Among them, 12 stations were selected for assimilation and 12 stations were selected for verification in the Beijing-Tianjin-Hebei (JJJ), 24 stations were selected for assimilation and 24 stations were selected for verification in the Yangtze River delta (YRD), and 9 stations were selected for assimilation and 9 stations were selected for verification in the Pearl River delta (PRD).

We have added some explanations in Lines 361-370, and in lines 489-492.

**9 Line 371: Why are hourly concentrations above 800 μg m-3 considered unrealistic? Hasn't had China intense pollution events where this limit was exceeded in terms of hourly concentration? In any case, this should be argued much better if observations are removed. Also, why are observations where the departure of the ensemble mean of the first guess exceeds 100 μg m-3 removed?**

In Schwartz et al. (2012), $PM_{2.5}$ values $> 200$ μg m$^{-3}$ were deemed unrealistic and were not assimilated. And observations leading to innovations exceeding 100 μg m$^{-3}$ were also omitted. Considering that China has had intense pollution events, $PM_{2.5}$ values larger than 800 μg m$^{-3}$ were discarded in this work. Also observations leading to innovations exceeding 100 μg m$^{-3}$ were also omitted. The statics show that 8 observations were discarded because they were larger than 800 μg m$^{-3}$ and 243 (around 1.5%) were discarded due to leading to innovations exceeding 100 μg m$^{-3}$.

We have added some explanations in Lines 386 and 389.

**10 Line 408: What is the impact of considering that no correlations exist between emission variables. What is the impact on the assimilation and the forecast?**

The emissions variables are related to each other. The correlations between the variables were reduced when perturbing the emissions without considering the correlations. Thus, the chemical forecast would deviate from the truth to some degree. Fortunately, the perturbed emissions were only used in the initialization and spin-up experiment and expC. Therefore, there were no impact on expJ and the control run except for expC.

We have added some explanations in Lines 770-775.

**11 Lines 460-461: What is it, are the emissions perturbed or not in expC? According to this line not, but according to the statement in lines 450-452, the emissions are perturbed by adding random noise.**

The emissions were perturbed by adding random noise in expC.

They were the prescribed emissions $E_t^p$ themselves in the control experiment.

So they were not perturbed. Lines 460-461 described the emissions in control experiment and they were right.

**12 Lines 566-570: Where are the numbers in this paragraph coming from?**

**Please explain and present them.**

Figure 6 shows time series of emission scaling factors extracted from the ensemble mean of the analyzed $\lambda_{NO}^a$, $\lambda_{SO2}^a$ and $\lambda_{NH3}^a$.

**13 Line 609: Replace "analysing" with "analysis".**

We have replaced "analysing" with "analysis" in Line 632.

**14 Line 649: What exactly is "dramatic"? How large is that? Please replace.**

We replaced "dramatic" with "very large" in Line 672.

**15 Lines 1097-1101: Authors should specify if the analysis presented in the**

**figures include all verification stations in each region or only some of them. In**

**addition, authors should also clarify to which dates the analysis presented in the**

**figures corresponds.**

All these verification stations presented in figure 1 in each region are used to calculate the statistics from 6 to 16 October. And we have clarified in Figure 10.

[revised manuscript text omitted]

---

## Author Response (AR3)

Feb. 28, 2017.

*Atmos. Chem. Phys.*

RE: Manuscript Number: acp-2016-732

Dear Editors:

Thank you very much for your kind decision letter on our paper entitled "**Improving**

**PM$_{2.5}$ forecast over China by the joint adjustment of initial conditions and source**

**emissions with an ensemble Kalman**" (acp-2016-732). We are grateful for the helpful comments from you and the reviewer. We have changed the manuscript according to his suggestions. We are looking forward to hearing from you soon.

Sincerely Yours,

Zhen Peng

**Response to Reviewer #3's comments:**

We thank Reviewer # 3 for his thoughtful comments and suggestions that have helped to improve this manuscript. Our responses to comments (in bold style) and the corresponding changes to the manuscript are detailed below.

**I gave went through the manuscript and the authors made a decent effort to try to address the issues I raised in my comments. I still have one comment though and that is that the authors should be more honest with their results. I'm not saying they're hiding information or twisting it around but they should acknowledge that for some cases the assimilation just didn't improve things, I'm talking about the performance of the assimilation in JJJ. There is some improvement in the first 24 hrs but then there is no difference between the control run and the forecasts of both experiments. Considering they're focusing on the 48hr forecast this is not a minor issue. The authors do mention this in section 5.4 but then in the discussions and conclusions they state "Large improvements were achieved for almost all the 48-h forecasts, particularly in the YRD and PRD. However, relatively smaller improvements were achieved in the first 24-h forecast in the JJJ region, which …". They focus on the 48hrs in the analysis of the results and then switch to the 24 hr forecast in the conclusions because it did have some improvement. I can understand that the authors want to highlight the things that worked in their assimilation, especially after all the work they have done, which is really good. All I'm saying is that the authors should be honest with the results and if something doesn't work just say so, it doesn't make the work less valuable. Based on the results the authors provide, the assimilating simply doesn't work for the 48 hr forecast in some regions.**

We realized that DA did work for some regions in certain periods although it cannot work perfectly for everywhere at all the time and it shows that there is no difference between the control run and the forecasts of both experiments for the second day forecast in JJJ. So I have replaced "relatively smaller improvements were achieved in the first 24-h forecast in the JJJ region" with "it did show some improvements in the first 24-h but then there is no difference between the control run and the forecasts in the JJJ region afterwards" in Lines 764-766.

Also I have added this in the abstract in Line 28 and in section 5.4 in Lines 739.

**What I also find interesting and the authors didn't address is that based on**

**the analysis runs (table 1) there is not much difference in assimilating IC or**

**IC+emissions (the authors themselves claim the statistics are the same, I would**

**say they are even better for expC in some cases). Anyway, considering that and**

**some of the results in figure 10 (in JJJ and YRD during the night), the**

**improvement of assimilating emissions in addition to IC seems to be not so big or**

**even negligible. I would have expected the authors say something about that as**

**well. Is there really a benefit of assimilating both IC and emissions everywhere?**

**my answer would be no, but the authors suggest that it is the case, at least imply**

**it.**

No. Actually the performances of assimilating emission are better in some regions.

From Figure 10 we can see that sometimes the performance of expJ was much better than expC in PRD and YRD. We addressed this in Lines 29-32, 722-725,

769-771.

**Last thing, one argument used to explain the performance of the system**

**there was the sparsity of the network, but in PRD the network is even sparser**

**and the system performs well so I don't see the argument. The authors should**

**better explain why in JJJ it is a problem but not in PRD.**

In this work, our interested regions were JJJ (36-40 °N, 110-120 °E), YRD (27-35 °

N, 116-122 °E) and PRD (22-24 °N, 112-115 °E). There were 12, 24 and 9 stations used for the $PM_{2.5}$ assimilation experiment in JJJ, YRD and PRD respectively. From figure 1 we can see that the measurement coverage in JJJ was much sparser than that in the YRD or PRD. So we think that the sparser coverage of measurements in JJJ

may be one of the reasons.

Some specific comments (based on the version with the modifications highlighted):

**Line 705: it says P25, i suppose it should be PM25**

In this work, we used $P_{25}$ (the unspectiated aerosol contributions) to calculate $\lambda^f_{PM2.5}$, so it was right in Line 325. We have no special reasons for this. It is also fine to use $PM_{25}$ to calculate $\lambda^f_{PM2.5}$.

**Line 748-749: What do they mean with "Since we did not know the exact station type,….", Do they mean urban or rural? please reformulate or clarify. In that same sentence the "We" after the , should be "we".**

We have changed this sentence as "Since we did not know the exact observation environment of the sites, …." in Lines 367-368. Also we have replaced "We" with "we".

**Lines 1061-1066: As far as I understand the reasons they are providing go in the different direction as what they try to justify. They are addressing the similar performance of the control run and both forecast experiments in YRD during the night. The claim that the prior emissions are larger than the optimized emission during the night and therefore the control run performs worse during night and better during day. What I see in figure 10 is the opposite, the control run performs better during the night than during day. The RMS and bias is smaller during the night than during the day. I would strongly suggest the authors to revise this part.**

In this work, the 48-h forecasts were performed at each 0000 UTC from 6 to 16 October 2014 with the hourly forecast output for both assimilation experiments. So in Figure 10, daytime starts from 1~11 UTC and 25~35 UTC (corresponding to 9-19 Local Time). So it was right in the text and we added the forecasts time in Line 1233. Also we labeled the daytime in Figure 10.

**Lines 1099-1102: I would ask the authors to reformulate these lines, I'm sure it can be improved.**

We have changed these sentences in Lines 727-736.

**Line 1104: replace "RRD" with "PRD"**

We have replaced "RRD" with "PRD" in Lines 738.

[revised manuscript text omitted]